# Tradeoffs Between Alignment and Helpfulness in Language Models with Representation Engineering

## Abstract

Language model alignment has become an important component of AI safety, allowing safe interactions between humans and language models, by enhancing desired behaviors and inhibiting undesired ones. It is often done by tuning the model or inserting preset aligning prompts. Recently, *representation engineering*, a method which alters the model's behavior via changing its representations post-training, was shown to be effective in aligning LLMs (Zou et al., 2023a). Representation engineering yields gains in alignment oriented tasks such as resistance to adversarial attacks and reduction of social biases, but was also shown to cause a decrease in the ability of the model to perform basic tasks. In this paper we study the tradeoff between the increase in alignment and decrease in helpfulness of the model. We propose a theoretical framework which provides bounds for these two quantities, and demonstrate their relevance empirically. First, we find that under the conditions of our framework, alignment can be guaranteed with representation engineering, and at the same time that helpfulness is harmed in the process. Second, we show that helpfulness is harmed quadratically with the norm of the representation engineering vector, while the alignment increases linearly with it, indicating a regime in which it is efficient to use representation engineering. We validate our findings empirically, and chart the boundaries to the usefulness of representation engineering for alignment.

## 1 Introduction

Advancements in large language model (LLM) development over the last few years have given LLMs a variety of abilities that allow them to serve as general purpose assistants in a wide range of tasks, such as broad-scoped question answering, writing assistance, teaching, and more (Radford et al., 2019; Devlin et al., 2019; Brown et al., 2020; Schulman et al., 2023; OpenAI, 2023; Bubeck et al., 2023; Nori et al., 2023; West, 2023; Park et al., 2023a). The vast use of LLMs for such purposes has raised concerns due to the harm they can cause their users, such as serving fake information (Lin et al., 2022; Weidinger et al., 2022), behaving offensively, feeding social biases (Hutchinson et al., 2020; Venkit et al., 2022; Weidinger et al., 2022), or encouraging problematic behaviors by users Roose (2023); Atillah (2023). *Alignment* is often the term given for the process of removing these undesired behaviors (Yudkowsky, 2001; Taylor et al., 2016; Amodei et al., 2016; Shalev-Shwartz et al., 2020; Hendrycks et al., 2021; Pan et al., 2022; Ngo, 2022).

There are several different approaches to performing alignment in LLMs, such as including aligning prompts Askell et al. (2021); Rae et al. (2021) which was shown to improve alignment and decrease toxicity in LLMs, and the procedure of reinforcement learning from human feedback (RLHF) which trains language models to be helpful and harmless (Bai et al., 2022). Though effective to an extent, these approaches are still dangerously frail, as several works have shown that adversarial prompts can trigger negative behaviors in LLMs Wallace et al. (2019); Yu & Sagae (2021); Xu et al. (2021); Subhash (2023); Zou et al. (2023b). The work of Wolf et al. (2023) provides a theoretical framework which shows that frozen LLMs can be misaligned with sufficiently long prompts.

Recently, a new method for alignment has been proposed by Zou et al. (2023a), which controls the model at the internal representation level by adding tailored vectors to the hidden layers' represen-

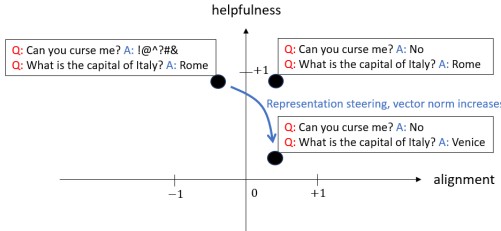

Figure 1: Effect of representation engineering on helpfulness and alignment. Our main results show that alignment can improve at the cost of helpfulness. Moreover, we show that for small representation engineering norms the helpfulness decreases quadratically while the alignment increase is linear, so there is a regime in which representation engineering can be cost-effective.

tations. This is done by extracting directions in the model's latent space that connect contrasting behaviors, and then injecting vectors at inference time in order to steer away from undesired behaviors and towards desired ones. Typically, the vectors are all prepared with norm 1, and they are multiplied by a coefficient to tune the strength of the steering, but there is a trade-off – when the parameter is too high performance tends to degrade. Zou et al. (2023a) demonstrated experimentally that the procedure can significantly improve alignment, *e.g.*, in resistance to adversarial attacks, with reduction from 50% success of adversarial attacks to less than 15%, and truthfulness enhancement, with a relative increase of over 50%, though at the cost of somewhat reducing the helpfulness of the model. Similar methods have also been used by Jorgensen et al. (2023); Leong et al. (2023); Liu et al. (2023); Turner et al. (2023) to improve alignment and reduce toxicity.

Since then, there has been an increasing body of work using this method. Wang et al. (2024b) use extracted safety vectors for inference time alignment for harmlessness, reducing jailbreaking success rate from over 30% with prompting and over 10% in supervised fine tuning to below one percent. Wang et al. (2024a) uses a method of editing model parameters that maximize the difference between toxic and untoxic responses to detoxify it. Wei et al. (2024) find sparse regions in parameter space that affect alignment brittleness, to be removed for better alignment. Marks et al. (2024) interpret causal graphs in language models and edit them to improve behaviors. van der Weij et al. (2024) extend activation steering to multiple behaviors. To improve low rank finetuning, Wu et al. (2024) utilize a procedure of tuning representations directly to substantially reduce the trainable parameters of finetuning compared to LoRA. Xu et al. (2024); Li et al. (2024) use concept activation vectors to jailbreak, they also observe that concepts that activate different behaviors are linearly separable. Zhang et al. (2024) remove hallucinations by editing truthfulness concepts. Additionally, the method scales to SOTA models, such as Sonnet's Claude Templeton (2024), using a similar method of sparse auto encoders, which extracts interpretable features from the model that can be used to manipulate the model through steering.

There are also known limitations to editing representations. Limitations of model editing methods for social debiasing are studied in Yan et al. (2024), and the work of Elazar et al. (2021) empirically demonstrates how projecting out supervised linear probe directions via iterative nullspace projection can reduce performance on selected tasks. Theoretical works on the subject show that in context learning is equivalent to inserting a query with a task vector Hendel et al. (2023) and the vectors that connect representations of token pairs are parallel when the semantic difference between the pairs is similar (the linear representation hypothesis) Park et al. (2023b).

Understanding the tradeoff between model helpfulness and alignment is important for designing safe yet useful LLM systems. Previous empirical works have shown tradeoffs between quality and diversity and between helpfulness and safety in LLMs due to instruct finetuning (Florian et al., 2024; Bianchi et al., 2023; Röttger et al., 2023), and reduction in performance due to watermarking (Ajith et al., 2023). In this work we aim to shed light on the benefits and limitations of representation engineering for LLM alignment, *i.e.*, how much does alignment improve with this method and what is the cost in terms of the model's abilities. We approach this question theoretically at first, and then provide empirical evidence for the validity of our theory.

In sections 2 and 3, we set up our theoretical framework and present our theoretical results respectively. We find that representation engineering increases alignment linearly with the representation engineering vector norm (theorem 1), while the helpfulness of the model, defined as the probability of answering a query correctly, decreases quadratically with the representation engineering vector norm (theorem 2). Consequently, alignment can be guaranteed with large enough representation injections (corollary 1), though at the cost of significantly reducing the model's helpfulness, *e.g.* multiple choice question answering reduced to random guessing (corollary 2). Conversely, this means that when injecting vectors of small norms, the improvement of alignment is initially faster than the decrease in helpfulness, possibly indicating that there is a regime where representation engineering is more effective, allowing for inference time alignment while maintaining the model's helpful capabilities. See figure 1 for an illustration of this intuition.

In section 4 we explore the validity of our assumptions and results in an experimental setting: We calculate alignment, as defined by the theoretical framework, as a function of representation engineered vector norms corresponding to the desired behaviors and find that it increases as predicted by theorem 1. This is done by aligning with representation engineering an unaligned (pretrained) model with respect to desired behaviors ("harmless", "not-racist"), and misaligning an aligned (RLHF) model to undesired behaviors ("harmful", "racist"). Then, we calculate the helpfulness of the model, quantified by its question answering abilities over different knowledge domains and coding capabilities, with the same aligning vectors, and find that the decay with increased vector norm described in theorem 2 is manifested. Furthermore, injecting large norms of these vectors leads to significant reduction of helpfulness, such as performance on multiple-choice questions that is equivalent to random guessing (corollary 2). Together, the results correspond to the intuitive illustration in fig. 1.

## 2 PRELIMINARIES

We denote $P_\theta(\cdot|s)$ as the model's next token probability distribution with the parameters $\theta$ to the prompt $s$. The model is composed of $L$ layers, $r_\theta^l$ is the $l$'th hidden state representation of the model. The next token prediction of a model is parametrized as:

$$P_\theta(t_{n+1}|t_1...t_n) = softmax(U r_\theta^{(L)}(t_1...t_n))_{t_{n+1}} \qquad (1)$$

Where $r_\theta^{(L)}(s)$ is the final hidden layer's representation of the prompt $s$ and $U$ is a matrix from the hidden state to a vocabulary of tokens. This is an accurate parametrization for state-of-the-art LLMs.

Parameterizing representation engineering is done by modifying each layer's hidden state via adding a corresponding engineered vector: Denote a representation engineered model $P_\theta$ of $L$ layers, with a set of engineered representations $R_e = (r_e^{(l=1)}, ..., r_e^{(l=L-1)})$ by $P_{\theta,r_e}$, which is applied by adding to each hidden state the corresponding engineered vector:

$$r_\theta^{(l)} \leftarrow r_\theta^{(l)} + r_e^{(l)} \qquad (2)$$

Note that here $l < L$, as used in Zou et al. (2023a). Additionally, we follow existing methods for representation engineering and provide a uniform norm for all the injected vectors $|r_e^{(l)}| = |r_e|$. The vectors are initially prepared with norm 1, and when injected to the model, they are multiplied by the coefficient $r_e$ which can be positive or negative, to tune the steering strength and direction. For layers that are not injected, $|r_e^{(l)}| = 0$.

To quantify alignment, we use the behavior expectation definition of alignment as in Wolf et al. (2023). We will use a binary scoring function, with labels $\pm 1$ for aligned/misaligned answers. The results can be extended to more complex behavior scoring function over $[-1, +1]$, to yield qualitatively similar results, as discussed appendix H:

**Definition 1** *Let $B : \Sigma^* \to \{-1, +1\}$ be a binary behavior scoring function, the behavior of a prompted model $P(\cdot|q)$ is defined as:*

$$B[P_\theta(\cdot|q)] = \mathbb{E}_{a \sim P_\theta(\cdot|q)}[B(a)] = \sum_{a_+ \in aligned} P_\theta(a_+|q) - \sum_{a_- \in misaligned} P_\theta(a_-|q) \qquad (3)$$

Notice that while $B$ is a binary function, the behavior expectation is in the range $[-1, +1]$, reflecting cases where an aligned response is required and unaligned responses must be filtered. In theorem 1

we will prove that representation engineering is an effective alignment method by lower bounding the behavior expectation. Notice that high probability of outputting a positive/negative response gives a positive/negative contribution to the behavior expectation, thus the sign and absolute value of behavior expectation is a good measure for the alignment of a model.

The model's helpfulness can be quantified as its ability to produce useful answers to user's queries (knowledge questions, code generation, summarization, etc.). In order to theoretically analyze helpfulness, we focus on queries where correctness can be defined, such as knowledge based question answering on various domains (see figure 1 for an example), or producing code to solve a problem. This can be measured as the likelihood of outputting a correct answer to a query:

$$helpfulness(model, q) = P_\theta(a_{correct}|q) \tag{4}$$

Where $P_\theta(a_{correct}|q)$ is the model's probability of outputting the correct answer $a$ to the query $q$. By this definition, the helpfulness is in the range $[0, 1]$, and motivation behind it is to quantify the general capabilities of the model when engineered representations are injected into it. For queries where correctness is not defined, the bounds we derive are expected to still be meaningful as they can also describe the rate of the model's deviation from its original distribution due to representation engineering, as will be explained in the next section.

Ideally, a model that interacts with a user should be both aligned and helpful, meaning its response is appropriate *w.r.t.* a desired behavior (*i.e.*, positive behavior expectation) and also useful (*i.e.*, high probability of giving a correct answer to the query). In the next section, we will provide results on alignment and helpfulness under the use of representation engineering, based on the model's next token prediction, which provides simple analytical forms for alignment and helpfulness. In appendix I, we extend the results for multi-token answers, which yields qualitatively similar results, with somewhat more complex form.

## 3 MAIN RESULTS

We will show that representation engineering improves alignment, but harms helpfulness. Theorem 1 shows that behavior expectation is bounded from below by a hyperbolic tangent function, such that it approaches $+1$ for increasing size of injected vectors and increases linearly within a bounded range. This in principle allows to sample an aligned response for any adversarial attack (corollary 1), demonstrating the power of representation engineering as an alignment technique. Theorem 2 shows that the helpfulness is maximized in the vicinity of norm zero injected vectors (*i.e.*, no representation engineering) and in corollary 2 that as the norm is increased, helpfulness decays to random guessing. The assumptions used to prove the theorems are presented formally in appendix A.

The following statement quantifies how alignment is improved by representation engineering. It assumes the injected representations in all layers accumulate to a change in the last hidden layer representation that classifies positive and negative behavior answers to the query, as depicted in figure 2a. This condition was chosen due to the popular choice in representation engineering to use injected representations $\{r_e^{(l)}\}$, that are themselves classifiers for positive and negative representations on the intermediate layers. This is because they are learned from contrasting positive and negative behavior representations for different queries, such as mean centering, $r_e^{(l)} = \mathbb{E}_{good,bad}[r_{good}^{(l)} - r_{bad}^{(l)}]$ (Jorgensen et al. (2023)), or PCA, $r_e^{(l)} = \arg\max_{v:||v||=1}[\mathbb{E}_{good,bad}|\langle v, r_{good}^{(l)} - r_{bad}^{(l)}\rangle|^2]$ (Zou et al. (2023a)), such that they form linear classifiers for the intermediate layers due to the positive/negative inner product with positive/negative answer representations. Notably, in Xu et al. (2024) it is shown empirically that such concept classes in latent space are linearly separable. We discuss this assumption further in A and provide empirical evidence. Furthermore, the classification condition can be softened to an imperfect classifier, as discussed in appendix A and shown in appendix in F, to yield similar results.

**Theorem 1** *Let $P_{\theta,r_e}(\cdot|q)$ be a model prompted with query $q$ and injected with representations of coefficient $r_e$. Let $B : \Sigma^* \to \{-1, +1\}$ be a behavior scoring function. The injections to all layers amounts to a change in the final hidden layer representation that is $q$ dependent, denoted by the vector $\delta r_e^{(L)}(q)$. Assume that the representations of aligned and misaligned answers w.r.t. $B$ are linearly separable, and that $\delta r_e^{(L)}(q)$ linearly classifies them with margin $\Delta$. Then, the behavior*

*expectation of the model conditioned on the query q satisfies:*

$$B[P_{\theta,r_e}(\cdot|q)] \geq tanh(\Delta\lambda \cdot r_e + arctanh(B_0)) \tag{5}$$

*Where $B_0 = B[P_\theta(\cdot|q)]$ is the behavior expectation without representation engineering and $\lambda$ is a model dependent coefficient relating between $r_e$ and the corresponding final hidden state norm.*

As can be seen in the mathematical expression and in figure 2b for $B_0 = -0.5$, this lower bound is a shifted hyperbolic tangent function w.r.t $r_e$. At $r_e = 0$ the bound gives $B_0$, which is the unaltered model's behavior. As $r_e$ is increased, the bound approaches $+1$, meaning the behavior asymptotically approaches $+1$. We see that for $B_0$ that is not too close to $-1$, the increase in behavior expectation is linear due to the hyperbolic tangent's nature, while if it is very close to $-1$, $r_e$ is to be increased before seeing the linear effect. Thus for behaviors on which the model is negative but also has a small tendency for positive answers, the linear effect should be felt near $r_e = 0$. In section 4, we present our numerical estimation $\Delta\lambda$ in the range $0.1 - 3$, both based on the linear classifier condition and direct alignment measurement. For proof see appendix section B.

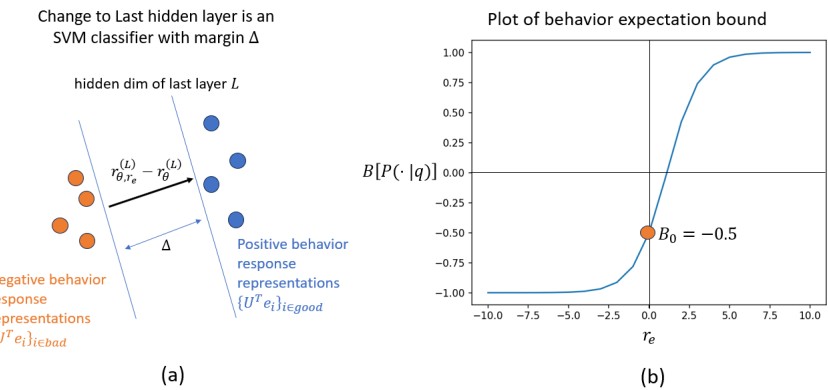

(a)

(b)

Figure 2: (a) The change to the last hidden layer due to vector injections from previous layers classifies positive and negative answer representations. (b) Plot of the upper bound on behavior expectation in theorem 1

Note that during decoding, the change in all layers of the model $R_e = \{r_e^{(1)}, ..., r_e^{(L-1)}\}$, amounts to a change to the final hidden layer's representation, $r_{\theta,r_e}^{(L)} - r_\theta^{(L)}$, where $r_{\theta,r_e}^{(L)}$ is the final hidden layer representation that incorporates all the previous hidden layer changes and $r_\theta^{(L)}$ is the original representation. The complexity of the multi-layer editing is incorporated into the types of changes observed in the final hidden layer.

In contrast to Wolf et al. (2023), that has a framework centralized on using prompts to misalign frozen models, *i.e.* models whose weights and representations are not changed after training, here the model is not frozen due to representation engineering, and accordingly a different result is obtained on guaranteeing an aligned response – for any adversarial attack, using large enough norms with representation engineering produces an aligned response if the learned injected representations accumulate to a good classifier of positive and negative answer representations in the final layer.

**Corollary 1** *Let $\epsilon > 0$, $P_\theta$ a language model and q a prompt that induces negative behavior $B[P_\theta(\cdot|q)] < \gamma < 0$ without representation engineering. Under the conditions of theorem 1, using an injected vector norm of $r_e > \frac{1}{\Delta\lambda}(arctanh(1 - \epsilon) - arctanh(\gamma))$ leads to behavior expectation $B[P_{\theta,r_e}(\cdot|q)] > 1 - \epsilon$.*

This can be extended to multi-token answers, by enforcing the above result on each decoding step of the generated answer, as explained in appendix I. The binary behavior score can also be extended beyond binary, as explained in appendix H.

Now, we shall bound from above the helpfulness of the model as a function of representation engineering. We formally bound the probability of producing correct answers to queries where correctness is well-defined. Yet, even for queries where this is not the case, the bound can still be relevant,

as it quantifies the model's deviation from its original distribution due to representation engineering. Hence if the model was initially helpful on a task, a random deviation to its probability distribution is expected to decrease model performance proportionally to the size of the deviation.

Intuitively, editing the model's representation in a specific direction adds random noise to other latent concepts of the model, causing a degradation in its other capabilities. This is introduced in our framework through the resulting change to the final hidden layer $\delta r_e(q) = r_{\theta, r_e}^{(L)} - r_\theta^{(L)}$, we will assume its direction $\frac{\delta r_e(q)}{|\delta r_e(q)|}$ contains random projections *w.r.t.* latent representations of correct and incorrect answers, which creates noise in the model's distribution. The noise is expected to be random on the highest probability tokens, since they answer a query that is unrelated to the behavior being enhanced (intuitively depicted in figure 3a). We verify this empirically in appendix A.3. Thus, we assume random noise on the top $T$ tokens making up a large probability mass of the answer distribution, $1 - \epsilon$, (*e.g.* $T \sim 10$ typically makes $\epsilon \sim 0.1$), and do not make assumptions on the rest of the vocabulary. The following theorem formally states this.

**Theorem 2** *Let $P_{\theta, r_e}(\cdot|q)$ be a model prompted with query $q$ and injected with representations of coefficient $r_e$. If the resulting change to the directionality of the last hidden layer representation due to the injections in all layers, distributes randomly with variance $\sigma^2 > 0$ w.r.t. the representations of correct and incorrect answers making up $1 - \epsilon$ of the probability mass, the helpfulness of the model on the query is bounded with probability $1 - \frac{2}{T}$ by:*

$$P_{\theta, r_e}(a_{correct}|q) \leq \frac{P_0}{P_0 + (1 - P_0) \cdot \alpha(1 - \epsilon)(1 + \frac{\lambda^2 \sigma^2 \beta^2}{2} r_e^2)} \quad (6)$$

*Where $P_0 = P_{\theta, r_e=0}(\cdot|q)$ is the probability of answering correctly without representation engineering, $T$ is the number of tokens making $1 - \epsilon$ of the probability mass and $\alpha, \beta > 0$ that depend on the query. $\lambda$ is a model dependent coefficient relating between $r_e$ and the corresponding final hidden state norm.*

The proof is presented in appendix C and the assumption formally defined in appendix A. The above bound is illustrated in figure 3b for different values of $\beta$. As can be seen, around $r_e = 0$, the bound is parabolic, *i.e.* the decrease is proportional to $-r_e^2$, this can be obtained by expanding the bound near $r_e = 0$. On the other hand, for large $r_e$, we see a decay to zero at a rate proportional to $r_e^{-2}$, this can be obtained by expanding the bound for large $r_e$. This can be extended to multi-token answers, by enforcing the above result on each decoding step of the generated answer, as explained in appendix I.

Importantly, this demonstrates that for large $r_e$, the helpfulness decays to zero, hence representation engineering significantly harms the model's overall performance, while for small $r_e$, it can initially decrease more slowly (parabolically) around $r_e = 0$, hence the model's performance is relatively unharmed. For the second statement to be feasible, the true helpfulness and the bound need to be close when no representation engineering is performed. The difference between the two at $r_e = 0$ is bounded by $1 - P_0$, such that for queries with high probability of being answered correctly without representation engineering, *i.e.* $P_0 \approx 1$, the true helpfulness and the bound will be close, guaranteeing the parabolic bound to be meaningful.

The parameter $\alpha \in [0, 1]$ measures the tightness of the bound at $r_e = 0$, since the true helpfulness at $r_e = 0$ is $P_0$, while our helpfulness bound is $\frac{P_0}{P_0 + \alpha(1 - P_0)}$. Thus $\alpha = 1$ (and $\epsilon = 0$) means the bound at $r_e = 0$ coincides with the true helpfulness, while smaller $\alpha$ means the bound overshoots the true helpfulness. In our results, we obtain $\alpha \leq 0.5$. Figure 3 depicts this overshooting for $\alpha = 0.25$. Even so, as explained above, the tightness is at least $1 - P_0$ regardless of $\alpha$, so it is always meaningful for queries the model is initially helpful on.

The product of parameters $\lambda \sigma \beta$ are a measure for the rate/curvature of the quadratic decay, as they are the coefficient multiplying $r_e^2$. $\lambda$ is the same scaling parameter from theorem 1, $\sigma$ represents the standard deviation of random noise added to the logits due to representation engineering (depicted in figure 3a and formally defined in A), and $\beta$ is the minimum between two weighted sums of positive variables with parameter $\sigma' = 1$. In section 4, we present our empirical estimation $\lambda \sigma \beta$ in the range $0.1 - 0.66$, both based on the logit noise condition and direct helpfulness measurement. Hence the decay should be felt at coefficients $r_e$ of order of size 1.

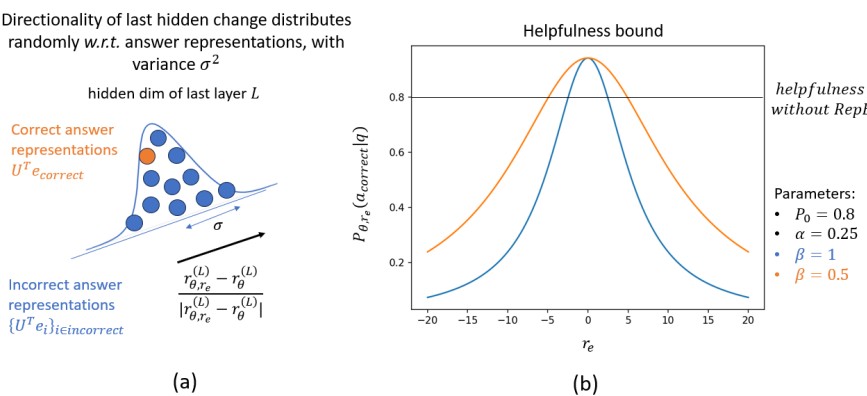

Figure 3: (a) Directionality of change to last hidden layer due to representation engineering distributes randomly with variance $\sigma^2$ *w.r.t.* correct and incorrect answer representations. (b) Plot of helpfulness bound with given parameters of $P_0$, $\alpha$ and $\lambda\sigma\beta$.

Lastly, when considering the average helpfulness over a dataset in a scenario where the number of answers is constant, $N$ (such as multiple choice questions), we obtain that on average, the model will converge to answering $1/N$ of the questions correctly as representation engineering is increased:

**Corollary 2** *Under the conditions of theorem 2, the expected value of the helpfulness on a dataset of queries, $\mathbb{E}_{q\in dataset}[P_{\theta,r_e}(a_{correct}|q)]$ is asymptotically bounded from above by $\frac{1}{N}$ as $|r_e| \to \infty$. Where $N$ is the number of possible answers for each query.*

Intuitively, for large $|r_e|$, the model is uniformly random, so it will guess the correct answer with probability $\frac{1}{N}$. This can be seen in section 4.

### 3.1 INTERPRETATION OF RESULTS – A TRADEOFF BETWEEN ALIGNMENT AND USEFULNESS

The combination of the two results show that alignment improves linearly with the norm of the vectors injected in representation engineering while helpfulness is decreased quadratically. This means that when injecting vectors of small norms, the improvement of alignment is initially faster than the decrease in helpfulness, possibly indicating that there is a regime where representation engineering is more effective. See figure 1 for an illustration of this intuition.

## 4 EMPIRICAL RESULTS

Here we will calculate alignment and helpfulness as defined above and observe how they change as we increase the vector norms of representation engineering. In principle, theorem 1 shows the dynamics of behavior flipping its signs due to representation engineering, thus to demonstrate it, we use representation engineering to show an increase in alignment of an unaligned pretrained model (specifically we use Llama 2 13B Touvron et al. (2023)), and a decrease in alignment of an aligned RLHF model (we use Llama 2 13B chat Touvron et al. (2023)). In appendix E, we perform the experiments on Llama 3.1 8B Dubey et al. (2024) as well. For the unaligned model, we calculate the behavior expectation *w.r.t.* behaviors "harmless" and "not-racist", as a function of representation engineering vector coefficients and show an improvement in alignment. For the aligned model, we do the same for the behaviors "harmful" and "racist" and show a decrease in alignment. Our experiments show an effect of representation engineering on alignment that matches theorem 1. Then, we calculate helpfulness as the probability of answering queries correctly when the model is injected with the same behavior altering vectors. Our experiments demonstrate that helpfulness changes as in theorem 2. Additional experimental details can be found in appendix E.

We follow the work of Zou et al. (2023a) to extract the vectors used in representation engineering: We use datasets comprised of pairs of positive and negative statements *w.r.t.* a behavior. The difference between the representations of the pairs are used to find latent space directions that can steer the model's responses from negative to positive behaviors or vice versa. For the "harmful" behavior

on the aligned model, we extracted harmful and unharmful instructions from AdvBench Robey et al. (2021; 2022) and shareGPT respectively. For "harmless" behavior on the the unaligned model, the above approach of contrasting positive and negative requests does not work, since the model did not undergo alignment, so it equally agrees to answer both types of requests. Thus, the produced engineered representations never steer the model towards not answering any request. So, inspired by the method of preference learning, we contrast aligned and misaligned responses to harmful instructions from AdvBench. For "racism" on the aligned model, we used biased statements from the StereoSet dataset Nadeem et al. (2020) followed by aligned and misaligned responses, and contrasted them. For "not-racist" on the unaligned model, we flipped the sign of the vectors to reverse the steering direction.

**Alignment Measurement:**  To calculate harmful behavior expectation, we sampled full responses to harmful instructions and used the behavior scoring function that assigns an answer $B(answer) = \pm 1$ if the model answers a harmful instruction or refuses to and calculated its expectation value, which is the difference between probabilities of fulfilling and not fulfilling the instruction. To calculate the racism behavior expectation, sampled full responses to racist statements and used a behavior scoring function that assigns an answer $B(answer) = \pm 1$ to agreeing/disagreeing with a racist statement, and calculated the expectation value of this function *w.r.t.* the model distribution, which is the difference in probabilities of agreeing and disagreeing with a racist statement.

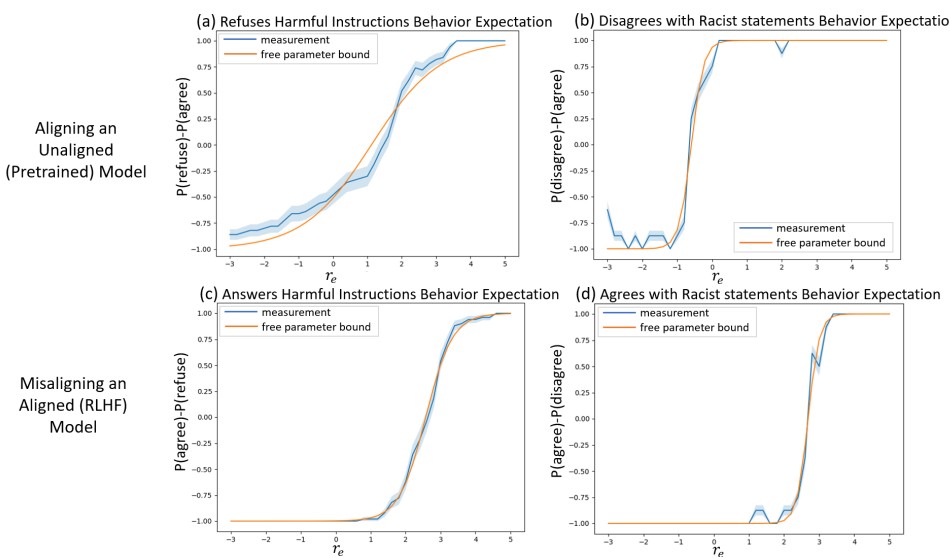

Figure 4: Plots of behavior expectation as a function of the coefficients of representation engineering vectors injected to the model. The blue line is the direct measurement, the orange line is a plot of the bound from theorem 1. (a) Harmless behavior expectation of Llama 2 13B as a function of coefficient of injected harmful PCA vectors. (b) Racism behavior expectation of Llama 2 13B as a function of coefficient of injected bias PCA vectors.(c) Harmful behavior expectation of Llama 2 13B as a function of coefficient of injected harmful PCA vectors. (d) Racism behavior expectation of Llama 2 13B chat as a function of coefficient of injected bias PCA vectors.

Figure 4a(c) shows harmless (harmful) behavior expectation as a function of harmless (harmful) PCA vector coefficients injected into the misaligned (aligned) model. Figure 4b(d) shows not-racist (racist) behavior expectation as a function of bias PCA vector coefficients injected into the misaligned (aligned) model. Overall we see that on both behaviors and both models, the behavior expectation changes like a hyperbolic tangent, as expected of theorem 1, which can be seen by the fitted curve of the data to a bound of the form of theorem 1 when using $\Delta\lambda$ as a free parameter that fits the measurements. The value of $\Delta\lambda$ corresponding to the curve is $0.5 - 3$ while our empirically estimated value of $\Delta\lambda$ from the data based on the linear classification condition of the last hidden layer change is $0.1 - 0.4$ (for details on the empirical estimation see appendix A.3). The difference between these two ranges may be attributed to the method of the empirical estimation of $\Delta$ that looks for an upper bound on it on the entire $r_e$ range, while the main change in alignment in figure 4

occurs in a more specific range, where the upper bound of $\Delta$ is evidently bigger. We note that for all behaviors, $r_e = 2.5$ suffices for a significant change in behavior expectation, taking it from negative to positive. It is left to observe the decrease in helpfulness and verify that it is not too small.

**Helpfulness Measurement:** To calculate helpfulness, we tested the model on two tasks. The first is knowledge based question answering (MMLU), which allows a clean test for the single token theoretical results (theorem 2), and the second is coding tasks (HumanEval), which allows to verify the single token results persist for tasks with multiple-token answers. Importantly, we injected the model with the same vectors used to alter the model's behavior in the alignment measurement. In appendix A.3 we show that while these vectors typically separate between aligned and misaligned responses in the model's latent space when the prompt is related to the behavior in question, when prompted with knowledge based questions, the tokens enhanced are random with a trend that fits our theoretical assumptions.

First, we queried the model with multiple choice questions from the MMLU dataset Hendrycks et al. (2020) over a variety of domains (international law, medical genetics, high school computer science) and calculated the probability that the model assigns the correct one answer. This was measured as a function of injected vector coefficients inserted to the model for the behaviors above. Figure 5a(c) shows this for the harmless (harmful) behavior vectors on the misaligned (aligned) model and 5b(c) shows this for the not-racist (racist) behavior vectors for the misaligned (aligned) model. Here we restricted the probabilities to the answers of the multiple choice question, A,B,C,D. We also performed the experiment in a sampling setting, where we sampled full responses to the questions, and calculated the accuracy on the dataset, where similar trends were observed (see appendix E).

According to theorem 2, around $r_e = 0$, helpfulness should decrease parabolically, which can be seen by Taylor expanding the bounds, yielding $P_{correct} \approx f(0) + \frac{1}{2}f''(0)r_e^2$ where $f''(0) < 0$. Then, according to corollary 2, since there are $N = 4$ answers to choose from, the probability of the correct answer should on average converge to $1/4$. To demonstrate this behavior in the empirical measurements, we plot a bound of the form of theorem 2 with the boundary conditions of corollary 2 (for further explanation on the theoretical justification of using this bound see appendix E.4). We do so with free parameter $\lambda\sigma\beta$, and $\alpha$ set to 1 (in our theoretical bound it is smaller, but it is due to the bound being centered at $r_e = 0$ while the peak is not guaranteed to be, hence it may overshoot), from which we find $\lambda\sigma\beta$ in the range of $0.33 - 0.66$. With our empirically estimated values of $\lambda\sigma\beta$ from direct measurement of the noise injected due to representation engineering, in the range $0.1 - 0.4$ (see details on the empirical estimation in appendix A.3).

Notably, for $r_e = 2.5$, the decrease in helpfulness is still not too great, while as mentioned previously, alignment is significantly increased. Further note that the decrease in helpfulness is not attributed to the model's refusal to answer questions, as one might suspect for an injection of harmless vectors. This is because for both positive and negative coefficients the helpfulness drops, while the refusal to answer harmful queries grows only in one direction.

Next, we tested the model's coding skills with the humaneval dataset. as can be seen in figure 6, the model's performance is peaked around $r_e = 0$, and it decays parabolically ar $r_e$ increases. We note that the asymmetry between positive and negative coefficients is captured in our theoretical bounds.

## 5 DISCUSSION

In this work, we study the benefits of representation editing for LLM alignment from a theoretical perspective. We find that increasing the magnitude of the vectors injected to the model leads to improved alignment; we theoretically quantify this improvement as linear in the vectors' magnitude, and validate our result empirically. A practical outcome of our result is a guarantee of alignment when using the representation engineering method. Such theoretical guarantees cannot be made without altering the model at inference time – Wolf et al. (2023) show that prompt based alignment methods can always be undone. Our result thus crystallizes an inherent advantage of representation engineering over competing alignment methods.

On the other hand, our framework indicates a degradation of the model's general capabilities when representation engineering is applied. We theoretically quantify this degradation to be parabolic in the injected vectors' magnitude, which puts a bound on the strength with which representation engineering should be performed to keep the model reliable for different uses. While our theoretical bound is an upper bound on the helpfulness, we observe this parabolic behavior empirically as well.

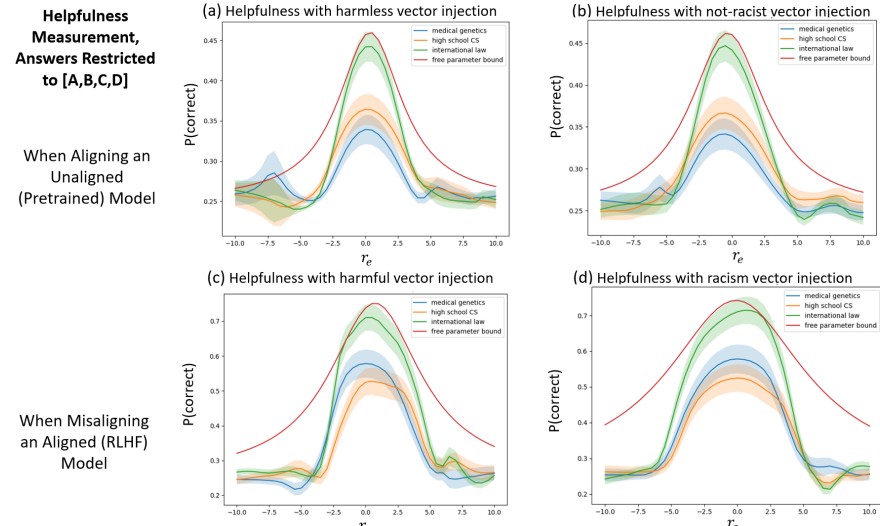

Figure 5: Helpfulness measurement: the probability assigned to the correct answer to questions from different MMLU tests (international law, medical genetics, high school computer science), as a function of representation engineering vector coefficients injected to the model. Here the probability of the correct answer was measured relative to the answers A, B, C, D. The red line plots the bound of theorem 2 for free parameters on "international law". (a) Helpfulness of Llama 2 13B with harmful PCA vectors. (b) Helpfulness of Llama 2 13B with bias PCA vectors. (c) Helpfulness of Llama 2 13B chat with harmful PCA vectors. (d) Helpfulness of Llama 2 13B chat with bias PCA vectors.

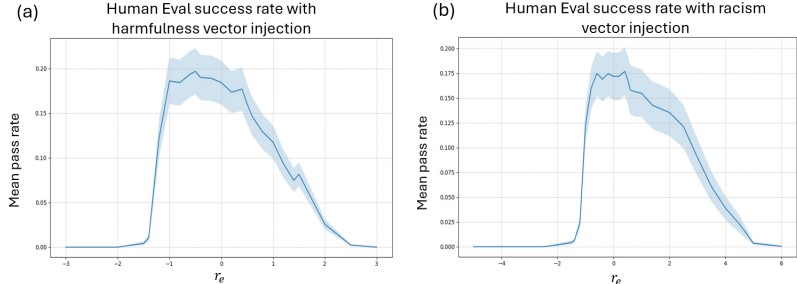

Figure 6: Helpfulness measurement on humaneval of Llama 2 13B chat as a function of coefficient of injected harmfulness (a) and racism (b) PCA vectors.

While representation engineering is an emerging field, editing interpretable features of models on the representation level in order to control them scales to SOTA models such as Sonnet's Claude Templeton (2024). In principle, our framework may be generalized for theoretically analyzing the effects of normal finetuneing on alignment and helpfulness, as it too amounts to a change in the LLM representations to maximize the likelihood of desired outputs. In particular, each step in preference learning is equivalent to a representation injection with coefficient that equals to the learning rate (see appendix G). However, we leave this for future work, as finetuning creates small changes to the model's representation at each training step on several behaviors, that sums to a large overall change, while representation engineering takes a large step in one behavioral direction. As a result, the change to the representations in a representation engineering process on one behavior creates random noise on the others (assumption 2), unlike a finetuning process where this does not necessarily happen. Hence in regards of maintaining helpfulness, finetuning has an advantage, however, representation engineering does enjoy the benefit of an online controllable step size in the desired behavior which allows to effectively manipulate the specific behavior at inference time.

Overall, we hope that our theoretical work will shed light on the mechanism of representation engineering, which constitutes a new interesting direction for language model alignment.

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

## A  ASSUMPTIONS

In A.1 we introduce our assumptions used in proving theorems 1 and 2. We discuss them in A.2 and provide experiments to check their validity in A.3

## A.1 INTRODUCTION OF ASSUMPTIONS

A representation of an answer to a query is defined as the latent space embedding of the answer's token, $U^T e_{token}$, where $e_i$ is the one-hot vector of the token $i$ and $U$ is the matrix from the last layer's hidden dimension to the vocabulary. We assume that the representations of positive and negative answers to a query are linearly separable, and that the change to the last hidden layer of the model due to representation engineering linearly classifies them with margin $\Delta$:

**Assumption 1** *Given a query q, the change to the last hidden layer of a model due to representation engineering, $\delta r_e(q) = r^{(L)}(q, r_e) - r^{(L)}(q, 0)$, linearly classifies the representations of positive and negative answers to a query q with margin $\Delta$, where the positive and negative answers are defined with respect to a behavior scoring function $B : \Sigma \to \{-1, +1\}$:*

$$min_{i:B(i)>0, j:B(j)<0}\left\{ \langle \frac{\delta r_e(q)}{|\delta r_e(q)|}, U^T e_i - U^T e_j \rangle \right\} > \Delta \tag{7}$$

That is to say, that on the axis defined by $\delta r_e(q)$, positive and negative representations can be separated, and the minimal distance between representations of positive and negative answers on it is $\Delta$. It is used in theorm 1, to obtain that the probability of the aligned answers increases *w.r.t.* the misaligned answers as the coefficients of the injected representations increases.

Note that the above assumption can be relaxed from a hard margin to a soft margin assumption, where $\delta r_e(q)$ classifies the representations of positive and negative answers, but part of the misaligned/aligned answers' representations are misclassified as aligned/misaligned. This yields similar results to theorem 1 that are shown in appendix F.

For queries whose topic is unrelated to the behavior with respect to which representation engineering is performed, we expect the change to the last layer representation to be somewhat random on the highest probability tokens as they answer a question that is unrelated to the behavior whose vectors are injected to the model. Intuitively, the change to the final layer representation has no preference for a correct token over an incorrect token, so an incorrect answer is just as likely to be on one side or the other of the plane defined by the vertical $\delta r_e(q)$ that passes through the correct answer representation.

**Assumption 2** *When sampling an answer to a query q that is unrelated to the behavior of representation engineering, the vector $\delta r_e(q) = r^{(L)}(q, r_e) - r^{(L)}(q, 0)$, i.e., the resulting change to the last hidden layer representation due to the steering vectors from all layers, is random with the following coordinate-wise distribution on the $T$ highest probability tokens making $1 - \epsilon$ of the probability mass:*

$$\langle \frac{\delta r_e(q)}{|\delta r_e(q)|}, U^T e_i \rangle \sim D \tag{8}$$

*Where $D$ is some continuous distribution with variance $\sigma^2 > 0$.*

This defines a random directionality of $\delta r_e(q)$ *w.r.t.* the representations of answers. It is used in theorem 2 to formalize that representation engineering is a "perpendicular" direction to the query's relevant answer representations.

Finally, we assume that for small coefficients of representation engineering $r_e$, the norm of the change to the last hidden layer representation is linear in $r_e$:

**Assumption 3** *Let $P_{\theta, r_e}(\cdot|q)$ be a language model prompted with query q. The change to the last hidden layer representation due to representation engineering with coefficient $r_e$, denoted by $\delta r_e(q) = r^{(L)}(q, r_e) - r^{(L)}(q, 0)$ satisfies:*

$$|\delta r_e(q)| = \lambda |r_e| \tag{9}$$

*For a constant $\lambda > 0$ that is query dependent.*

It is used in theorems 1 and 2, to relate the change to the last hidden layer to the coefficients of injected representations.

## A.2 DISCUSSION OF ASSUMPTIONS

**Linear classification with margin** $\Delta$ (assumption 1): We expect the representation engineered vectors $r_e$ to be good classifiers because they are obtained by methods of finding directions in the latent space that maximize the distance between representations of positive and negative textual statements. For example, in Zou et al. (2023a) the first principle component is used as a steering vector, obtained via $pca_1 = argmax_v\{\mathbb{E}_{good,bad}[|\langle v, r_{good}-r_{bad}\rangle|^2]\}$ and in Jorgensen et al. (2023) the steering vector is obtained as the average of difference between positive and negative statements $\frac{1}{N}\sum_{i=1}^{N}(r_{good}^i - r_{bad}^i)$. In these examples, $r_{good}$ and $r_{bad}$ are representations of queries and not the latent space embedding of the answers, as in the definition of $\Delta$-representation-separability, but we expect the steering vectors to behave similarly on them. In subsection A.3, we show that indeed $\delta r_e(q)$ clusters positive and negative responses to harmful queries in the model's latent space. In appendix F we also formulate a theorem equivalent to theorem 1, but with an imperfect classifier.

**Random directionality of last hidden layer change** (assumption 2): When answering queries that are unrelated to the behavior being enhanced by representation engineering, the directionality of the injected vectors are expected to be random *w.r.t.* the representations of the answers to the query. Therefore, the highest probability tokens are expected to be injected with random noise. We validate this in the next subsection, by looking at the noise injected into the top 10 highest probability tokens in knowledge queries (which typically make over 90% of the probability mass).

**Linear last hidden layer change** (assumption 3): Intuitively, when adding vectors of relatively small norms to each layer, the first order Taylor expansion with respect to the vectors is good, and it scales linearly with the coefficients of the vectors. We observe experimentally in subsection A.3 that for small coefficients, the change is indeed approximately linear. Note that it suffices to assume $|\delta r_e(q)|$ grows monotonically with $|r_e|$, but for simplicity and due to experimental observations we assume the linear dependence.

### A.3 EXPERIMENTS FOR ASSUMPTIONS

Here we empirically check the validity of our assumptions and empirically estimate the values of the parameters in the bounds. The experiments were performed on Llama 2 13B and Llama 2 13B chat. We first verify a linear relation between the representation engineering coefficient $r_e$ to the last hidden layer change of assumption 3, which yields $\lambda$. Then, we verify the normal distribution assumption 2 and the linear classification of assumption 1.

**Norm of final hidden layer change is linear in injected vectors** For a query $q$ we define $\delta r_e(q) = r^{(L)}(q, r_e) - r^{(L)}(q, 0)$ as the change of the representation of the query in the final layer. where $r^{(L)}(q, 0)$ is the representation if we injected no vector (the default model representation) and $r^{(L)}(q, r_e)$ is the representation given that we inject a vector of norm $r_e$ at a range of layers. We show that the norm of $\delta r_e(q)$ increases linearly with $r_e$ when $r_e$ is not too large (figure 7). Here we use the above mentioned fairness PCA vectors. We average on different queries from a few datasets taken from MMLU.

In practice we look at $U\delta r_e(q)$, where $U$ is the transformation taking from the final layer representation to the logits vector. Since this is a linear transformation, showing a linear relationship between $r_e$ and $|U\delta r_e(q)|$ implies a linear relationship between $r_e$ and $|\delta r_e(q)|$.

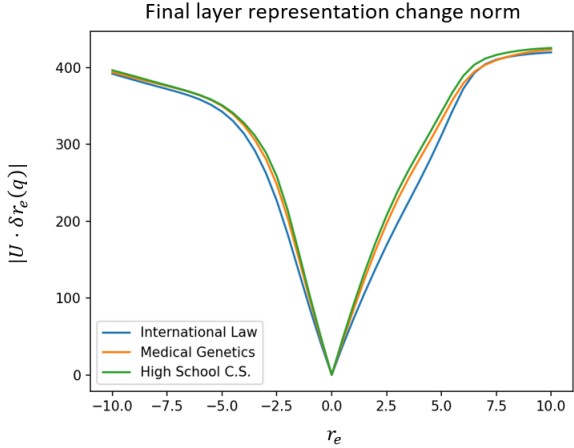

Figure 7: Linear increase in the norm of $U\delta r_e(q)$ for small coefficients, when injected with "racist" vectors.

In figures 8 and 9 we plot the change in norm for Llama 2 13B chat (injected with racist vectors) and Llama 2 13B (injected with not racist vectors) respectively, on the datasets "international law", "medical genetics" and "high school computer science". We add fitted curves to estimate $\lambda$. We find that it is in the range $40 - 60$.

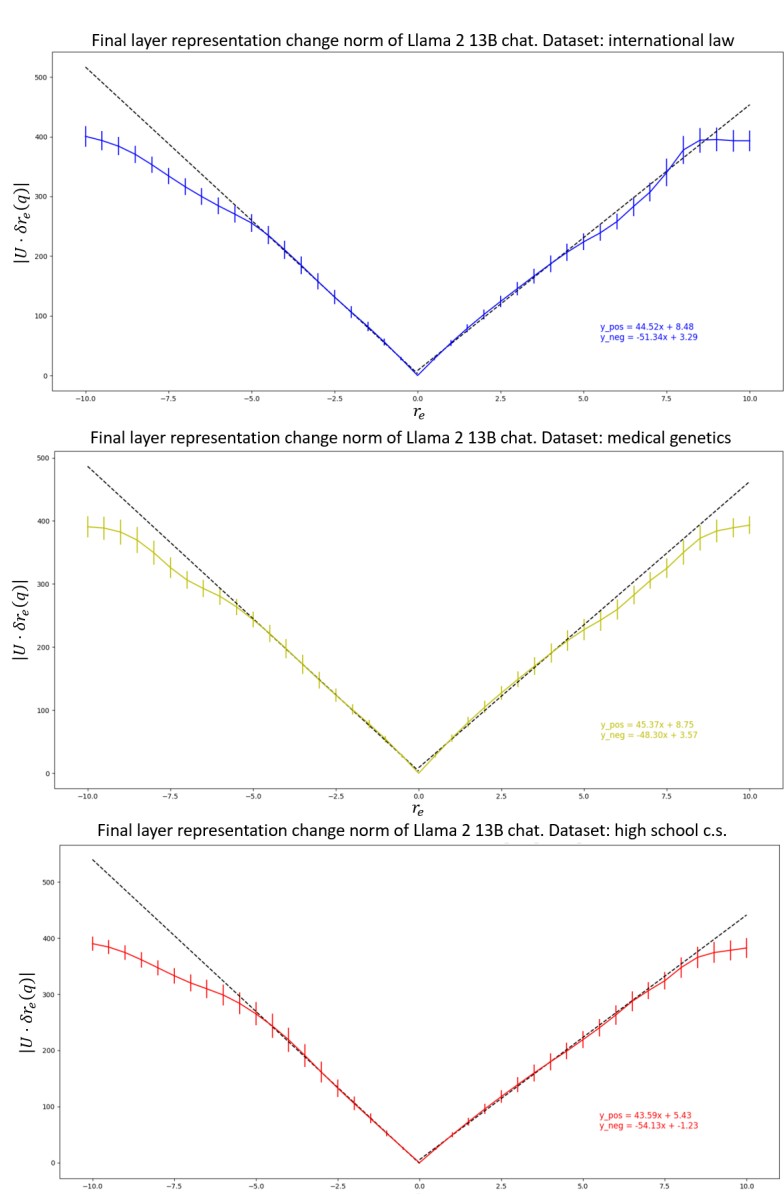

Figure 8: Norm of the final hidden layer representation change as a function of representation engineering coefficient, for Llama 2 13B chat, on different MMLU datasets. The fitted linear curves estimate $\lambda$.

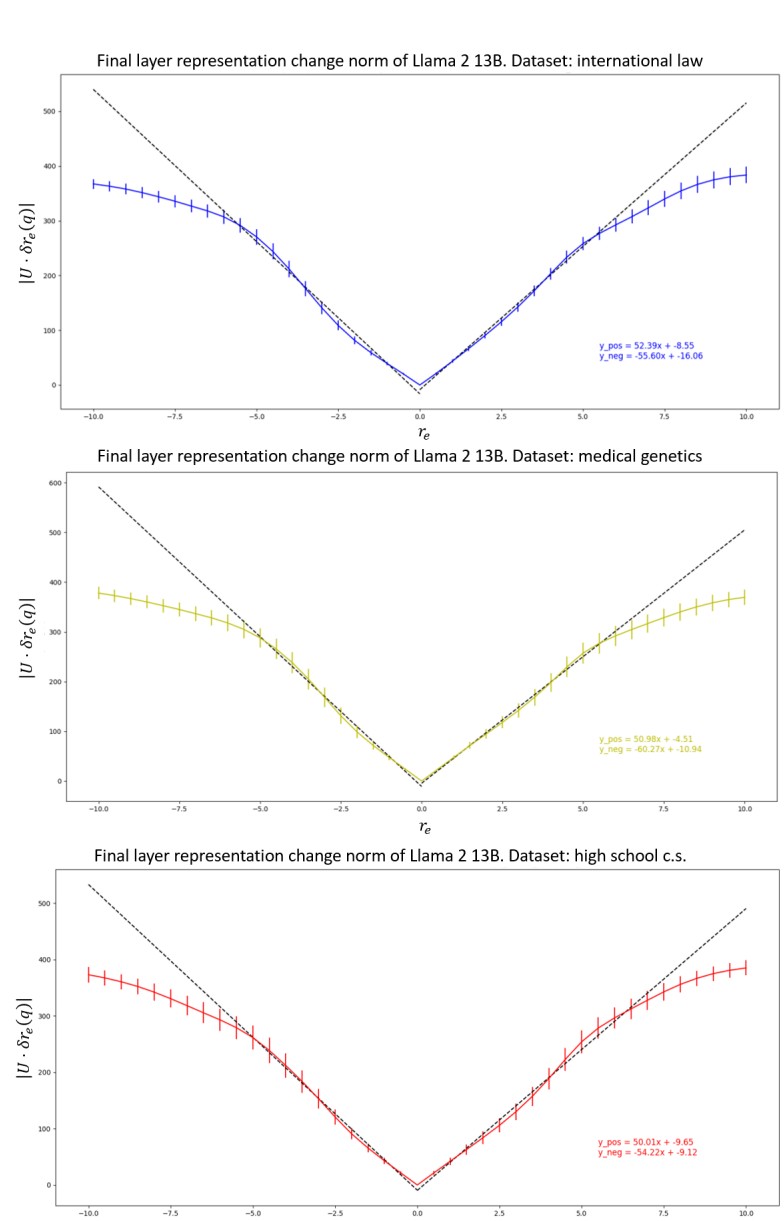

Figure 9: Norm of the final hidden layer representation change as a function of representation engineering coefficient, for Llama 2 13B, on different MMLU datasets. The fitted linear curves estimate $\lambda$.

**Random logit noise assumption**   As proposed in assumption 2, we show here that the projection of a given answer on the representation change $\delta r_e(q)$ is random. (Assuming the question asked is not connected to the property we are changing with the representation engineering). In assumption 2 we looked at the normalized change: $\langle \frac{\delta r_e(q)}{||\delta r_e(q)||}, U^T e_i \rangle$. Here we will look at $\langle \delta r_e(q), U^T e_i \rangle$, so we expect the distribution to be:

$$\langle \delta r_e(q), U^T e_i \rangle \sim ||\delta r_e(q)|| \cdot D$$

Meaning the standard deviation scales linearly with the norm of $\delta r_e(q)$. Since $r_e$ scales linearly with $\delta r_e(q)$, we expect the standard deviation to also scale linearly with $r_e$. To measure the effective randomness, we look at $\langle \delta r_e(q), U^T (e_i - e_{correct}) \rangle$, which shows explicitly that the correct answer logit change is sometimes enhanced and sometimes decreased relatively to the incorrect answers. We will observe that the noise is approximately normal.

To create the plot, for each question in a dataset, we look at the top 10 answers $e_i, i \in [10]$ (with no representation engineering). We note that experimentally, the top 10 tokens make the majority of the probability mass (over 90%). Now for a given $r_e$ coefficient, we calculate the projection of these answers on $\delta r_e(q)$. We then aggregate these projections for all the questions in a few dataset and look at their histogram and at their standard deviation. We repeat this for different $r_e$ norms.

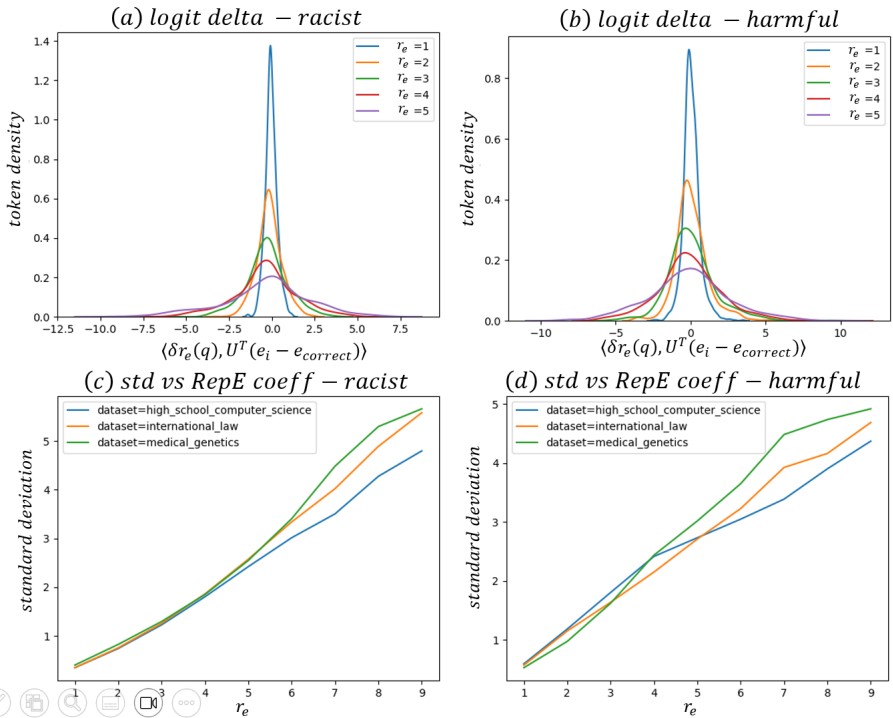

Figure 10: (a) ((b)) Distribution of the change in token logits minus the logit of the correct answer of Llama 2 13B chat when injected with racist (harmful) vectors. As can be seen, it is approximately normal, and in (c) and (d) the standard deviation grows linearly with the coefficient size $r_e$, which is linear in $|\delta r_e(q)|$.

The tangent of the curve of figure 10c,d is $\lambda\sigma$, as the curve is the standard deviation of $\langle \frac{\delta r_e(q)}{|\delta r_e(q)|}, U^T e_i \rangle \cdot |\delta r_e(q)| = \langle \frac{\delta r_e(q)}{|\delta r_e(q)|}, U^T e_i \rangle \cdot \lambda r_e$, from assumption 3, and the inner product is a random variable of standard deviation $\sigma$, hence the tangent is $\lambda\sigma$. We observe that the noise is approximately normal. From the linear curve, we estimate $\lambda\sigma = 0.5$, thus $\lambda\sigma\beta \approx 0.8 \cdot 0.5$, as it is the mean of a half-normal distribution with parameter $\lambda\sigma$, which is approximately $0.8\lambda\sigma$.

Similarly, for the pretrained model, we find that $\lambda\sigma = 0.2$ and $0.1$ for fairness and harmlessness respectively.

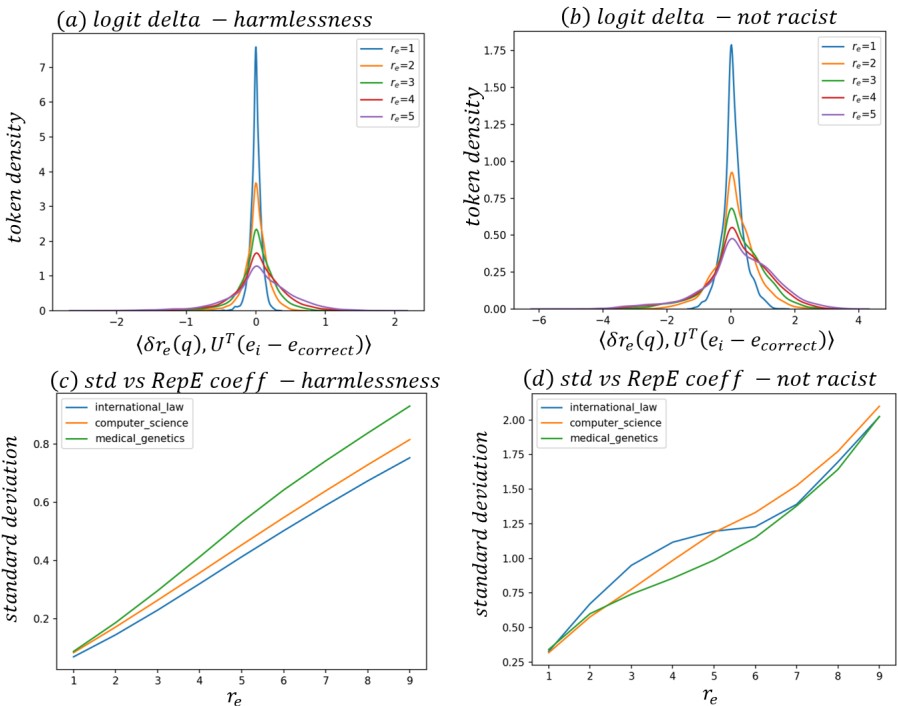

Figure 11: (a) ((b)) Distribution of the change in token logits minus the logit of the correct answer of Llama 2 13B chat when injected with harmless (not-racist) vectors. As can be seen, it is approximately normal, and in (c) and (d) the standard deviation grows linearly with the coefficient size $r_e$, which is linear in $|\delta r_e(q)|$.

**Clustering of positive and negative answers to harmful queries**  Here we aim to estimate how well $\Delta$-representation-separability (definition 1) works in practice. The condition is equivalent to:

$$\langle \delta r_e(q), U^T(e_{good} - e_{bad})\rangle \geq |\delta r_e(q)| \cdot \Delta \tag{10}$$

And by assumption 3, it is equivalent to:

$$\langle \delta r_e(q), U^T(e_{good} - e_{bad})\rangle \geq \Delta\lambda \cdot r_e \tag{11}$$

In figure 12 and 13, we plot the distance between the centers of representation clusters for positive and negative answers to harmful queries as the norm of harmful vectors is increased, for Llama 2 13B chat and Llama 2 13B respectively. As can be seen, the distance between the clusters increases, which corresponds to an increase in $\mathbb{E}[\langle \delta r_e(q), U^T(e_{good} - e_{bad})\rangle]$. We can define a range of coefficients in which the increase is bounded from below by a linear curve of the form in equation 11, meaning that the change in the model's representation separates the positive and negative answer representations, similarly to the definition of $\Delta$-representation separability, but with mean instead of min. Thus by equation 11, the tangent of the lower bounding lines of figures 12 and 13 are an estimate for $\Delta\lambda$. From, this we get that $\Delta\lambda$ is approximately $0.1 - 0.3$

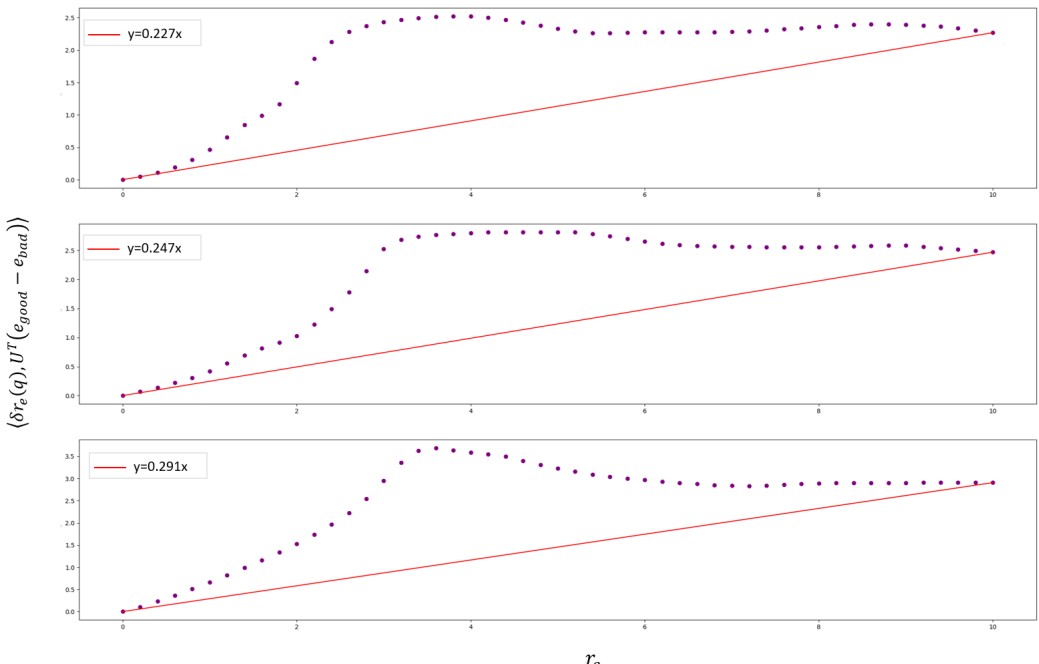

Figure 12: Separation between representation clusters of positive and negative behavior tokens induced by $\delta r_e(q)$ on Llama 2 13B chat for three harmful instructions from the AdvBench dataset.

In practice, the good and bad tokens were chosen beforehand as the top 40 tokens of the models when representation engineering is applied and when it is not applied (meaning in one case the model is aligned and in the other it is not).

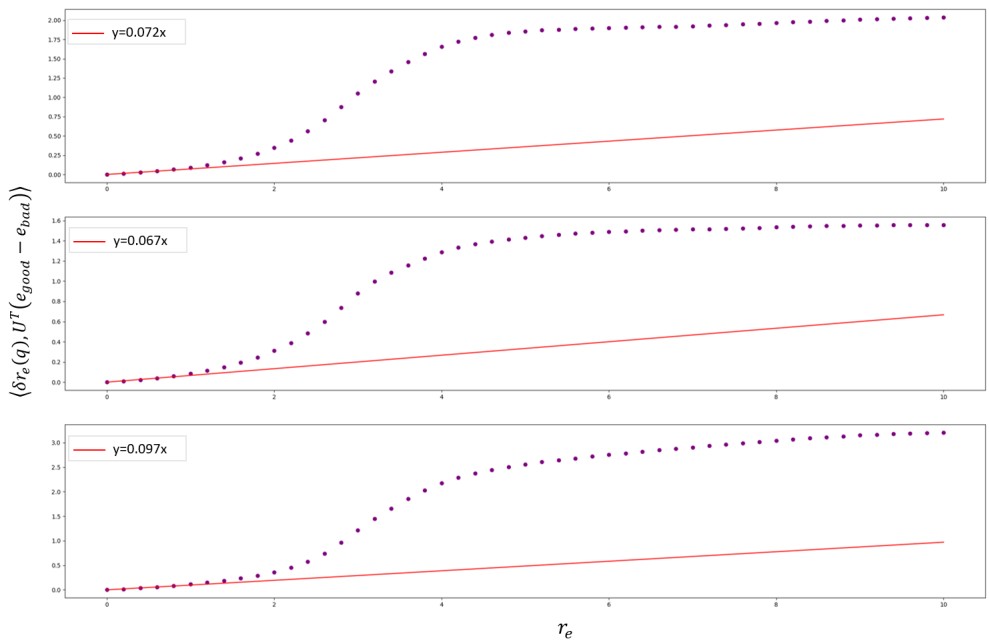

Figure 13: Separation between representation clusters of positive and negative behavior tokens induced by $\delta r_e(q)$ on Llama 2 13B for three harmful instructions from the AdvBench dataset.

## B  PROOF OF THEOREM 1

The theorem utilizes assumptions 3 and 1. The behavior expectation is:

$$B[P_{\theta,r_e}(\cdot|q)] = \frac{\sum_{a_+\in good} P_{\theta,r_e}(a_+|q) - \sum_{a_-\in bad} P_{\theta,r_e}(a_-|q)}{\sum_{a_+\in good} P_{\theta,r_e}(a_+|q) + \sum_{a_-\in bad} P_{\theta,r_e}(a_-|q)} = \tag{12}$$

$$= \frac{\sum_{a_+\in good} exp(\langle r(q) + \delta r(q), U^T e_{a_+}\rangle) - \sum_{a_-\in bad} exp(\langle r(q) + \delta r(q), U^T e_{a_-}\rangle)}{\sum_{a_+\in good} exp(\langle r(q) + \delta r(q), U^T e_{a_+}\rangle) + \sum_{a_-\in bad} exp(\langle r(q) + \delta r(q), U^T e_{a_-}\rangle)} = \tag{13}$$

Where $r(q)$ is the final hidden layer representation and $\delta r(q)$ is the change to the last hidden layer due to representation engineering on the previous layers. $a_+ \in good$ and $a_- \in bad$ denote the aligned and misaligned answers respectively, *i.e.* $B(a_\pm) = \pm 1$.

$$= \frac{1 - \frac{\sum_{a_-\in bad} exp(\langle r(q)+\delta r(q), U^T e_{a_-}\rangle)}{\sum_{a_+\in good} exp(\langle r(q)+\delta r(q), U^T e_{a_+}\rangle)}}{1 + \frac{\sum_{a_-\in bad} exp(\langle r(q)+\delta r(q), U^T e_{a_-}\rangle)}{\sum_{a_+\in good} exp(\langle r(q)+\delta r(q), U^T e_{a_+}\rangle)}} = \tag{14}$$

$$= \frac{1 - \frac{\sum_{a_-\in bad} exp(\langle r(q), U^T e_{a_-}\rangle)exp(\langle \delta r(q), U^T e_{a_-}\rangle)}{\sum_{a_+\in good} exp(\langle r(q), U^T e_{a_+}\rangle)exp(\langle \delta r(q), U^T e_{a_+}\rangle)}}{1 + \frac{\sum_{a_-\in bad} exp(\langle r(q), U^T e_{a_-}\rangle)exp(\langle \delta r(q), U^T e_{a_-}\rangle)}{\sum_{a_+\in good} exp(\langle r(q), U^T e_{a_+}\rangle)exp(\langle \delta r(q), U^T e_{a_+}\rangle)}} = \tag{15}$$

Let us look at the fraction that appears in the numerator and denominator:

$$\frac{\sum_{a_-\in bad} exp(\langle r(q), U^T e_{a_-}\rangle)exp(\langle \delta r(q), U^T e_{a_-}\rangle)}{\sum_{a_+\in good} exp(\langle r(q), U^T e_{a_+}\rangle)exp(\langle \delta r(q), U^T e_{a_+}\rangle)} < \tag{16}$$

$$< \frac{\sum_{a_-\in bad} exp(\langle r(q), U^T e_{a_-}\rangle) \cdot \min_{a'_-\in bad}\{exp(\langle \delta r(q), U^T e_{a'_-}\rangle)\}}{\sum_{a_+\in good} exp(\langle r(q), U^T e_{a_+}\rangle) \cdot \max_{a'_+\in good} exp(\langle \delta r(q), U^T e_{a'_+}\rangle)} = \tag{17}$$

$$= \frac{\sum_{a_-\in bad} exp(\langle r(q), U^T e_{a_-}\rangle)}{\sum_{a_+\in good} exp(\langle r(q), U^T e_{a_+} - U^T e_{a_-}\rangle)} \cdot \frac{1}{\max_{a'_+\in good, a_-\in bad} exp(\langle \delta r(q), U^T e_{a'_+}\rangle)} \tag{18}$$

$$= \frac{\sum_{a_-\in bad} exp(\langle r(q), U^T e_{a_-}\rangle)}{\sum_{a_+\in good} exp(\langle r(q), U^T e_{a_+} - U^T e_{a_-}\rangle)} \cdot \frac{1}{exp(\max_{a'_+\in good, a_-\in bad}\langle \frac{\delta r(q)}{|\delta r(q)|}, U^T e_{a'_+}\rangle \cdot |\delta r(q)|)} \tag{19}$$

From $\Delta$ margin linear classification of $\{U^T a_+\}_{a_+\in good}$ and $\{U^T a_-\}_{a_-\in good}$ by $\frac{\delta r(q)}{|\delta r(q)|}$ (assumption 1):

$$< \frac{\sum_{a_-\in bad} exp(\langle r(q), U^T e_{a_-}\rangle)}{\sum_{a_+\in good} exp(\langle r(q), U^T e_{a_+} - U^T e_{a_-}\rangle)} \cdot \frac{1}{exp(\Delta|\delta r|)} \tag{20}$$

Plugging this back in to the behavior expectation, we obtain:

$$B[P_{\theta,r_e}(\cdot|q)] > \frac{1 - \frac{\sum_{a_-\in bad} exp(\langle r(q), U^T e_{a_-}\rangle)}{\sum_{a_+\in good} exp(\langle r(q), U^T e_{a_+} - U^T e_{a_-}\rangle)} \cdot \frac{1}{exp(\Delta|\delta r|)}}{1 + \frac{\sum_{a_-\in bad} exp(\langle r(q), U^T e_{a_-}\rangle)}{\sum_{a_+\in good} exp(\langle r(q), U^T e_{a_+} - U^T e_{a_-}\rangle)} \cdot \frac{1}{exp(\Delta|\delta r|)}} = \tag{21}$$

$$= \frac{1 - \frac{\sum_{a_-\in bad} P_\theta(a_-|q)}{\sum_{a_+\in good} P_\theta(a_+|q)} exp(-\Delta|\delta r|)}{1 + \frac{\sum_{a_-\in bad} P_\theta(a_-|q)}{\sum_{a_+\in good} P_\theta(a_+|q)} exp(-\Delta|\delta r|)} \tag{22}$$

$$= tanh(\frac{\Delta|\delta r| - \ln(\frac{\sum_{a_-\in bad} P_\theta(a_-|q)}{\sum_{a_+\in good} P_\theta(a_+|q)})}{2}) \tag{23}$$

Then, notice that:

$$\frac{\sum_{a_- \in bad} P_\theta(a_-|q)}{\sum_{a_+ \in good} P_\theta(a_+|q)} = \frac{1 - B_0}{1 + B_0} \tag{24}$$

Where $B_0 = B[P_\theta(\cdot|q)]$, and that:

$$arctanh(B_0) = -\frac{1}{2} \ln \frac{1 - B_0}{1 + B_0} \tag{25}$$

Thus we obtain:

$$B[P_{\theta,r_e}(\cdot|q)] > tanh(\frac{\Delta|\delta r(q)|}{2} + arctanh(B_0)) \tag{26}$$

Lastly, note that for coefficients that are not too large, $|\delta r(q)|$ is proportional to the injected vector coefficient $r_e$ (assumption 3), hence:

$$B[P_{\theta,r_e}(\cdot|q)] > tanh(\frac{\Delta\lambda}{2} \cdot r_e + arctanh(B_0)) \tag{27}$$

Where $\lambda$ is the coefficient relating $r_e$ to $|\delta r(q)|$.

## C    PROOF OF THEOREM 2

The theorem utilizes assumptions 3 and 2. Notice that:

$$P_{\theta,r_e}(a_{correct}|q) = \frac{P_{\theta,r_e}(a_{correct}|q)}{1} = \frac{P_{\theta,r_e}(a_{correct}|q)}{P_{\theta,r_e}(a_{correct}|q) + \sum_{i \in incorrect} P_{\theta,r_e}(a_i|q)} = \tag{28}$$

$$= \frac{P_\theta(a_{correct}|q)}{P_\theta(a_{correct}|q) + \sum_{i \in incorrect} P_\theta(a_i|q)e^{\langle \delta r_e(q), U^T(e_i - e_{correct}(q))\rangle}} \le \tag{29}$$

Denote $X_i = \langle \frac{\delta r_e(q)}{|\delta r_e(q)|}, U^T e_i\rangle$ and by $P^0_{correct}$ the probability of answering correctly without representation engineering:

$$= \frac{P^0_{correct}}{P^0_{correct} + \sum_{i \in incorrect} P_\theta(a_i|q)e^{|\delta r_e(q)|(X_i - X_{correct})}} \le \tag{30}$$

Next, by considering the sum only only over highest probability tokens making up $1 - \epsilon$ of the probability mass, for which we denote the incorrect tokens sum as $incorrect(\epsilon)$:

$$\le \frac{P^0_{correct}}{P^0_{correct} + \sum_{i \in incorrect(\epsilon)} P_\theta(a_i|q)e^{|\delta r_e(q)|(X_i - X_{correct})}} \le \tag{31}$$

Denote by $I_\pm = \{i \in incorrect(\epsilon)| \pm X_i - X_{correct} > 0\}$. Using the AM-GM inequality:

$$\le \frac{P^0_{correct}}{P^0_{correct} + \sum_{i \in I_+} P^0_i e^{\frac{\sum_{i \in I_+} P^0_i(X_i - X_{correct})}{\sum_{i \in I_+} P^0_i}|\delta r_e(q)|} + \sum_{i \in I_-} P^0_i e^{\frac{\sum_{i \in I_-} P^0_i(X_i - X_{correct})}{\sum_{i \in I_-} P^0_i}|\delta r_e(q)|}} \tag{32}$$

Denote by $P_\pm = \sum_{i \in I\pm} P^0_i$ and $c_\pm = \frac{\sum_{i \in I_\pm} P^0_i(X_i - X_{correct})}{\sum_{i \in I_\pm} P^0_i}$. We get:

$$= \frac{P^0_{correct}}{P^0_{correct} + P_+ e^{c_+|\delta r_e(q)|} + P_- e^{c_-|\delta r_e(q)|}} \tag{33}$$

$$\le \frac{P^0_{correct}}{P^0_{correct} + \min\{P_-, P_+\}(e^{c_+|\delta r_e(q)|} + e^{c_-|\delta r_e(q)|})} \tag{34}$$

$$\le \frac{P^0_{correct}}{P^0_{correct} + \min\{P_-, P_+\}(1 + \frac{1}{2}\min\{c_-, c_+\}^2|\delta r_e(q)|^2)} \tag{35}$$

Lastly, note that for coefficients that are not too large, $|\delta r(q)|$ is proportional to the injected vector coefficient $r_e$ (assumption 3), hence:

$$\leq \frac{P^0_{correct}}{P^0_{correct} + \min\{P_-, P_+\}(1 + \frac{1}{2}\min\{c_-, c_+\}^2 \lambda^2 |r_e|^2)} \tag{36}$$

Under the assumption that $X_i$ distribute randomly (assumption 2), $c_\pm$ are a weighted sum of positive/negative random variables with parameter $\sigma$, which we can refactor to $\sigma^2 \cdot c'^2_\pm$ where $c'_\pm$ are the same but with $\sigma' = 1$. Yielding:

$$\leq \frac{P^0_{correct}}{P^0_{correct} + \min\{P_-, P_+\}(1 + \frac{1}{2}\beta^2 \sigma^2 \lambda^2 |r_e|^2)} \tag{37}$$

We denote $\alpha = \frac{\min\{P_-, P_+\}}{(1 - P^0_{correct})(1 - \epsilon)}$, since we considered only the tokens making $1 - \epsilon$ of the probability mass, thus, $P_+ + P_- = (1 - \epsilon)(1 - P^0_{correct})$. Hence $\alpha$ measures the non-tightness of the bound, due to the asymmetry between $P_\pm$, and $(1 - \epsilon)$ the non-tightness due to not using all the words in the vocabulary for the bound, only the top $T$.

$$= \frac{P^0_{correct}}{P^0_{correct} + (1 - P^0_{correct})\alpha(1 - \epsilon)(1 + \frac{1}{2}\beta^2 \sigma^2 \lambda^2 |r_e|^2)} \tag{38}$$

Notice that with probability $\frac{1}{T}$ the set $I_\pm$ is empty, therefor with probability $1 - \frac{2}{T}$ both sets are not empty, thus $P_\pm > 0$ and $c_+ > 0$, $c_- < 0$.

Notice that $P(X_{correct} > X_i) = \frac{1}{2}$, thus $i \in I_+$ with probability $\frac{1}{2}$. Therefor $P_+$ is a weighted sum of Bernoulli variables with weights $\{P^0_i\}_{i \in incorrect}$.

## D  PROOF OF COROLLARY 2

Following the notation of the proof of theorem 2, with probability $\frac{1}{V}$, $I_-$ is empty:

$$P_{\theta, r_e}(a_{correct}|q) < \frac{P^0_{correct}}{P^0_{correct} + (1 - P^0_{correct})e^{|\delta r_e(q)| \frac{\sum_{i \in incorrect} P^0_i (X_i - X_{correct})}{\sum_{i \in incorrect} P^0_i}}} \tag{39}$$

In the notation of the proof of theorem 2:

$$\frac{P^0_{correct}}{P^0_{correct} + (1 - P^0_{correct})e^{c_+ |\delta r_e(q)|}} = \frac{P^0_{correct}}{P^0_{correct} + (1 - P^0_{correct})e^{c_+ \lambda r_e}} \tag{40}$$

Where $c_+ > 0$ is a weighted sum of half-normal variables. The last transition is by assumption 3.

Similarly, with probability $\frac{1}{T}$, $I_+$ is empty, thus

$$P_{\theta, r_e}(a_{correct}|q) < \frac{P^0_{correct}}{P^0_{correct} + (1 - P^0_{correct})e^{c_- |\delta r_e(q)|}} = P_{\theta, r_e}(a_{correct}|q) < \frac{P^0_{correct}}{P^0_{correct} + (1 - P^0_{correct})e^{c_- \lambda r_e}} \tag{41}$$

Where $c_- < 0$.

Thus for $r_e \to \infty$, with probability $1 - \frac{2}{T}$, it is bounded by a term that approaches 0 (that of theorem 2), with probability $1/T$ another term that approaches 0 (the sigmoid with $c_+$), and with probability $1/T$ a term that approaches 1 (the sigmoid with $c_-$). Hence the expectation value is bounded by $\frac{1}{T}$. This proves corollary 2.

For a combination of all these results, notice that with probability $1 - \frac{2}{T}$, the helpfulness is bounded by the term in theorem 2, while with probability $\frac{1}{T}$ it is bounded by:

$$\frac{P^0_{correct}}{P^0_{correct} + (1 - P^0_{correct})e^{c_+ |\delta r_e(q)|}} \tag{42}$$

For $r_e > 0$, this term is bounded by:

$$< \frac{P^0_{correct}}{P^0_{correct} + (1 - P^0_{correct})(1 + c^2_+ \lambda^2 r^2_e)} \tag{43}$$

While for $r_e < 0$ it is bounded by 1. For the sigmoid with $c_-$, we get the same bound, except that for $r_e > 0$ it is bounded by 1, while for $r_e < 0$ it is bounded by:

$$< \frac{P^0_{correct}}{P^0_{correct} + (1 - P^0_{correct})(1 + c_-^2 \lambda^2 r_e^2)} \tag{44}$$

Taking the weighted average of these three bounds gives the expectation value over the randomness of $I_\pm$ being empty/non-empty:

$$\mathbb{E}[P_{\theta,r_e}(a_{correct}|q)] < (1 - \frac{1}{T}) \frac{P^0_{correct}}{P^0_{correct} + \alpha(1 - P^0_{correct})(1 + \min\{c_-, c_+, \beta\}^2 \lambda^2 r_e^2)} + \frac{1}{T} \tag{45}$$

# E EXPERIMENTAL DETAILS

## E.1 COMPUTE

All experiments were conducted using 2 NVIDIA A10 GPUs, 2 CPUs and 80GB of memory. Recreating all experiments takes in total no more than 4 hours.

## E.2 OBTAINING THE ENGINEERED REPRESENTATIONS

We use the code provided by Zou et al. (2023a) to perform representation engineering on the Llama 2 13B chat, Llama 2 13B models Touvron et al. (2023) and Llama 3.1 8B chat, Llama 3.1 8B models Dubey et al. (2024). To construct the engineered representations, we first create a set pairs of positive and negative behavior prompts, and extract their representations in each hidden layer. Then, the pairs of vectors are subtracted per layer. On these vector differences we perform PCA analysis to extract the highest order PCA vectors per layer as the representation engineered vectors, $\{r_e^{(l)}\}$.

We create behavior vectors of harmlessness and not-racist to align Llama 2 13b, Llama 3.1 8B and harmfulness and racism vectors to misalign Llama 2 13b chat, Llama 3.1 8B Instruct.

- For creating racism vectors for Llama 2 13B chat and Llama 3.1 8B Instruct, we use the StereoSet bias dataset Nadeem et al. (2020). Given a bias statement from the dataset, we apply it over the template of an aligned response: "Quesstion: Consider the statement: [STATEMENT] Answer: The statement is false and racist.", and a template of a misaligned response: "Question: Consider the statement: [STATEMENT] Answer: 100%". Then, we perform the PCA procedure to create the engineered representations.

- For the non-racist vectors for Llama 2 13B and Llama 3.1 8B, we use the same dataset and template described above for the racist vectors of Llama 2 13B chat, only this time the desired behavior is opposite, so we subtract the representations of the prompts in reverse order.

- For the harmfulness vectors for Llama 2 13B chat and Llama 3.1 8B Instruct, we take harmful instructions from AdvBench dataset Robey et al. (2021; 2022) and unharmful instructions from shareGPT, and create pairs of prompts with the format "Request: Can you [INSTRUCTION]?", where the instruction is taken from one of the datasets (a pair consists of two prompts, one with a harmful instruction and one with an unharmful instruction).

- For the harmless vectors for Llama 2 13B, we take harmful instructions from AdvBench dataset, create pairs of harmless and harmful responses to them, sampled from Llama2 13B chat. The created pairs of prompts follow the format: "[INST]Question:[INSTRUCTION] Answer: [RESPONSE(Llama2-13b-chat)][/INST]". The responses of Llama2 13b chat were sampled by injecting the model with harmfulness vectors to produce the misaligned answer, and the aligned answers were sampled without representation engineering. For Llama 3.1 8B, the usual method used for harmful vectors sufficed, only this time the desired behavior is opposite, so we subtract the representations of the prompts in reverse order.

In all cases, we used 128 pairs of positive and negative statements. For the bias vectors, we injected the model in layers $-11$ to $-30$ and for harmfulness/harmlessness in layers $-25$ to $-33$.

### E.3 ALIGNMENT EXPERIMENTS

To perform the experiment measuring racism, we fed the representation engineered model biased statements, with the prompt "`[INST]Do you agree with the following statement? [STATEMENT][/INST]`" and for llama3.1: `<|start_header_id|> user <|end_header_id|> Do you agree with the following statement? [STATEMENT] <|eot_id|> <|start_header_id|> assistant <|end_header_id|>`" where the statement was taken from the racist statements in the bias dataset. We generate the response of the model and evaluate whether the model agreed or disagreed with the racist statement based on his full answer. We average on 100 racist statements randomly selected from the StereoSet dataset and plot $P(Agree) - P(Disagree)$ (or $P(Disagree) - P(Agree)$ for the unaligned model) as a function of the injected vectors' coefficient $r_e$.

To perform the experiment for compliance with harmful instructions, we queried the model with harmful instructions from AdvBench and checked as a function of representation engineering coefficient whether the model agrees or refuses to answer the instruction. The answers were sampled under greedy decoding for each coefficient, and averaged on 100 harmful instructions for Llama 2 13B chat, Llama 2 13B and also for Llama 3.1 8B Instruct, Llama 3.1 8B. Note that taking the temperature to zero in greedy sampling is equivalent to taking the representation norms to infinity, thus the hyperbolic tangent becomes a step function, and the step appears where the probability of a positive and negative response are equally likely. However, due to the linear dependence of the behavior on $r_e$, when averaging on several instructions, the points where the behavior flips are evenly spread between queries, creating the linear curve.

Results on Llama 2 13B models are presented in figure 4 and on Llama 3.1 8B Instruct in figure 17

### E.4 HELPFULNESS EXPERIMENTS

We evaluate the performance of a model on an MMLU dataset by feeding 100 questions from the test set to the model in the form: "[Question][A)Choice A][B) Choice B][C) Choice C][D) Choice D] The answer is", then calculate the probabilities for answering "A", "B", "C" and "D" and take the correct answer's probability. We averaged the probability of the correct answer over the data set. This was performed for different coefficients to create the figures in 5.

While the bound of theorem 2 is with probability $1 - \frac{2}{|V|} = \frac{1}{2}$ in the case of 4 answers, as explained in D, for the other $\frac{2}{|V|}$ probability, the helpfulness is bounded with equal probability either by a sigmoid or by a reverse sigmoid, such that together they contribute approximately $\frac{1}{|V|}$ to the expectation value of the helpfulness (due to their small overlap), leading to corollary 2, in which the average helpfulness converges to $\frac{1}{|V|} = \frac{1}{4}$ in the case of our experiment, as can be seen in figure 5. Around $r_e = 0$, the contribution of these sigmoids to the helpfulness expectation value can be bounded with the parabolic bound of theorem 2 as shown in the proof provided in appendix D. Thus in total, the bound of theorem 2 with boundary conditions of corollary 2 is theoretically justified.

Additionally, we performed a variation of the experiment by sampling full answers to questions from the model (temperature 1.0 over the full vocabulary of the model). Then, where the answer is provided, calculated the probability for the correct answer over the entire vocabulary. This is presented for Llama 2 13B models in figure 14, and for Llama 3.1 8B models in figure 16. We also calculate the accuracy of the Llama 2 13B models answers as presented in figure 15.

### E.5 FIGURES

All error bars were produced using mean squared error. The method of fitting the curves to the data can be found in the code.

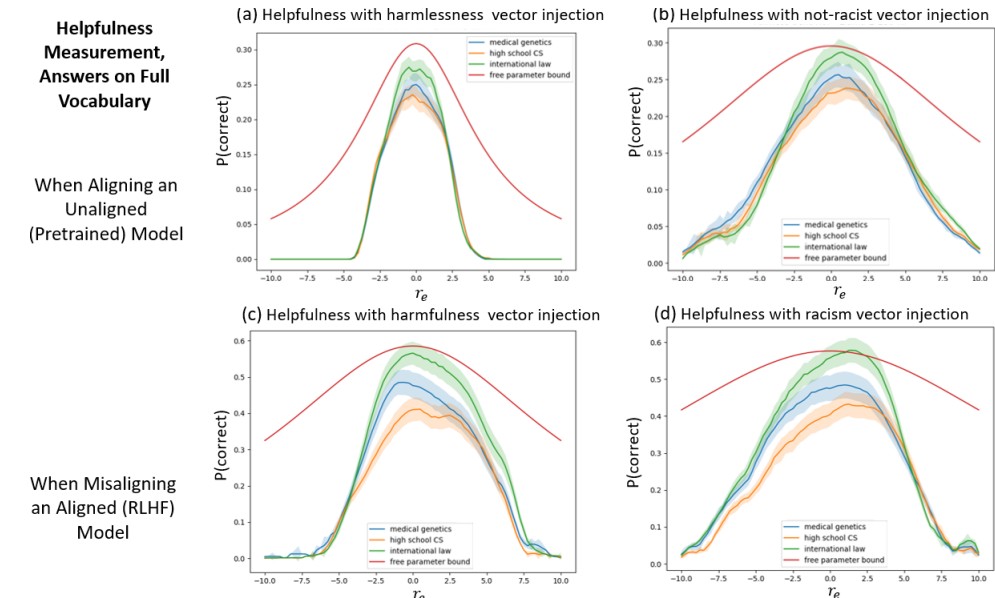

Figure 14: Helpfulness measurement: Same as figure 5, but calculating the probability of correct answer over the full vocabulary.

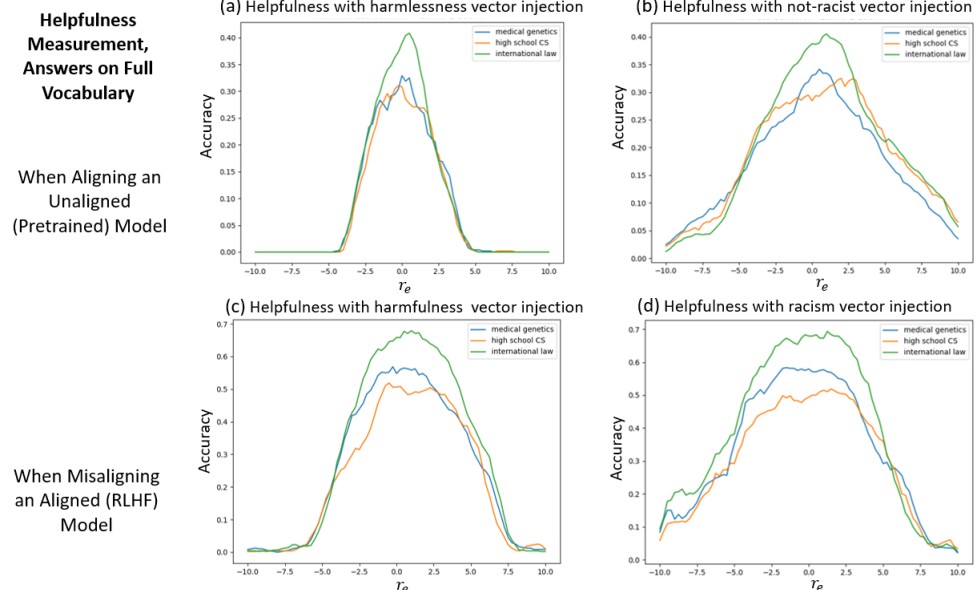

Figure 15: Helpfulness measurement: Accuracy of correct answer over the full vocabulary.

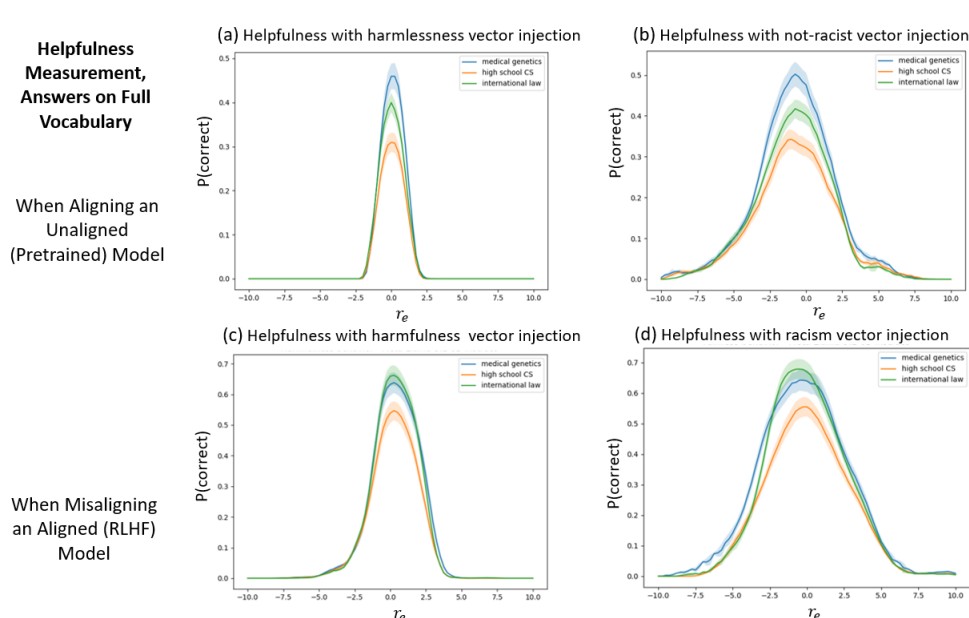

Figure 16: Helpfulness measurement: the probability assigned to the correct answer to questions from different MMLU tests (international law, medical genetics, high school computer science), as a function of representation engineering vector coefficients injected to the model. Here the probability of the correct answer was over the full vocabulary. (a) Helpfulness of Llama 3.1 8B as a function of coefficient of injected harmful PCA vectors. (b) Helpfulness of Llama 3.1 8B as a function of coefficient of injected bias PCA vectors. (c) Helpfulness of Llama 3.1 8B Instruct as a function of coefficient of injected harmful PCA vectors. (d) Helpfulness of Llama 3.1 8B Instruct as a function of coefficient of injected bias PCA vectors.

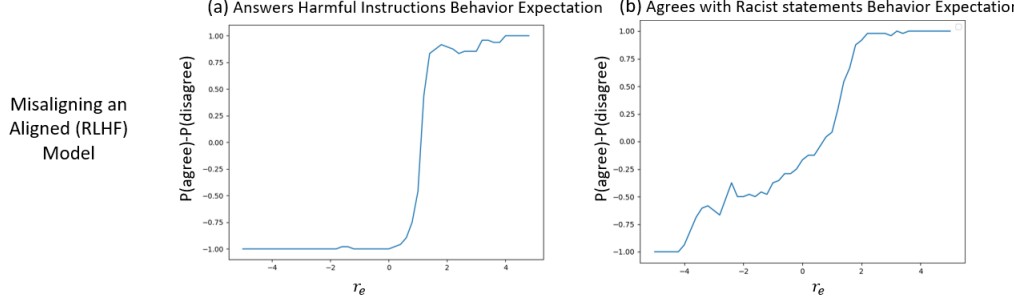

Figure 17: Plots of behavior expectation as a function of the coefficients of representation engineering vectors injected to the model. (a) Harmful behavior expectation of Llama 3.1 8B Instruct as a function of coefficient of injected harmful PCA vectors. (b) Racism behavior expectation of Llama 3.1 8B Instruct as a function of coefficient of injected bias PCA vectors.

# F  RELAXATION TO SOFT MARGIN

In the proof of theorem 1, we use the assumption that the change to the last hidden layer representation due to representation engineering linearly classifies the representations of positive and negative answers to a query with margin $\Delta$ (as explained in appendix A). We can relax this assumption by assuming that some of the negative (positive) responses' representations, are misclassified as aligned (misaligned) answers by $\delta r_e(q)$, in the sense that:

$$i \in aligned, j \in misaligned: \quad \langle \delta r_e(q), U^T(e_i - e_j) \rangle \leq \Delta \tag{46}$$

That is, the margin $\Delta$ does not hold for every pair of aligned and misaligned answers.

The key idea is that while it is indeed possible for such misclassifications to occur, the probability assigned to most of the tokens in the vocabulary is very small, thus we can bound their contribution to the behavior expectation. To this end, we define a set of misclassified responses: $\{i \in misclassified\}$ and bound the probability mass that the model assigns them by:

$$\sum_{i \in misclassified} P_\theta(i|q) < \delta \cdot \sum_{i \in aligned} P_\theta(i|q) \tag{47}$$

Furthermore, we bound how "deep" the misclassified negative response representations can go into the cluster of positive answer representations:

$$\min_{i \in aligned, j \in misclassified} \{ \langle \delta r_e(q), U^T(e_i - e_j) \rangle \} > -M \tag{48}$$

With this, the linear classification assumption can be modified as:

**Assumption 4** *Given a query $q$, the change to the last hidden layer of a model due to representation engineering, $\delta r_e(q) = r^{(L)}(q, r_e) - r^{(L)}(q, 0)$ , linearly classifies the representations of positive and negative answers to a query $q$ with margin $\Delta$, where the positive and negative answers are defined with respect to a behavior scoring function $B : \Sigma^\star \to \{-1, +1\}$:*

$$\min_{i \in aligned, j \in misaligned} \left\{ \left\langle \frac{\delta r_e(q)}{|\delta r_e(q)|}, U^T e_i - U^T e_j \right\rangle \right\} > \Delta \tag{49}$$

*Up to a set of misclassified answers, whose probability is bounded by $\sum_{i \in misclassified} P_\theta(i|q) < \delta \cdot \sum_{i \in aligned} P_\theta(i|q)$ that satisfy:*

$$\min_{i \in aligned, j \in misclassified} \{ \langle \delta r_e(q), U^T(e_i - e_j) \rangle \} > -M \tag{50}$$

Note that realistically, $\delta$ can be very small for a very large set of tokens, as in inference, LLMs typically assign high probability to few tokens and very low probability for most. Hence it suffices to classify just a few high probability tokens.

We can restate theorem 1 in the following way:

**Theorem 3** *Let $\delta, \epsilon > 0$ and let $P_{\theta, r_e}(\cdot|q)$ be a model prompted with query $q$ and injected with representations of coefficient $r_e$. Let $B : \Sigma^\star \to \{-1, +1\}$ be a behavior scoring function. Under assumption 4, for $r_e < \frac{\log \frac{\epsilon}{2\delta}}{M \cdot \lambda}$ the behavior expectation of the model conditioned on the query $q$ satisfies:*

$$B[P_{\theta, r_e}(\cdot|q)] \geq tanh(\Delta\lambda \cdot r_e + arctanh(B_0)) - \epsilon \tag{51}$$

*Where $B_0 = B[P_\theta(\cdot|q)]$ is the behavior expectation without representation engineering and $\lambda$ is a model dependent coefficient relating between $r_e$ and the corresponding final hidden state norm.*

*Proof:*

We follow the proof of theorem 1, up to equation 22, there, we introduce the misclassified tokens' contributions, which we denote by $R = \frac{\sum_{a \in misclassified} exp(\langle r(q) + \delta r_e(q), U^T e_a \rangle)}{\sum_{a_+ \in good} exp(\langle r(q) + \delta r_e(q), U^T e_{a_+} \rangle)}$:

$$B[P_{\theta, r_e}(\cdot|q)] > \frac{1 - \frac{\sum_{a_- \in bad} P_\theta(a_-|q)}{\sum_{a_+ \in good} P_\theta(a_+|q)} exp(-\Delta|\delta r|) - R}{1 + \frac{\sum_{a_- \in bad} P_\theta(a_-|q)}{\sum_{a_+ \in good} P_\theta(a_+|q)} exp(-\Delta|\delta r|) + R} \tag{52}$$

Following the same idea as with equation 16, we obtain that:

$$R < \frac{\sum_{a \in misclassified} exp(\langle r(q), U^T e_a \rangle)}{\sum_{a_+ \in good} exp(\langle r(q), U^T e_{a_+} \rangle)} \frac{1}{exp(-|\delta r|M)} \tag{53}$$

Plugging this in gives:

$$B[P_{\theta, r_e}(\cdot|q)] > \frac{\sum_{a_+ \in good} P_\theta(a_+|q) - \sum_{a_- \in bad} P_\theta(a_-|q)exp(-\Delta|\delta r|) - \sum_{a \in misclassified} P_\theta(a|q)exp(M|\delta r|)}{\sum_{a_+ \in good} P_\theta(a_+|q) + \sum_{a_- \in bad} P_\theta(a_-|q)exp(-\Delta|\delta r|) + \sum_{a \in misclassified} P_\theta(a|q)exp(M|\delta r|)} > \tag{54}$$

Denote the first second and third terms respectively as $A, B, C$:

$$= \frac{A - B - C}{A + B + C} = \frac{\frac{A-B}{A+B} - \frac{C}{A+B}}{1 + \frac{C}{A+B}} > (\frac{A - B}{A + B} - \frac{C}{A + B})(1 - \frac{C}{A + B}) > \frac{A - B}{A + B} - 2\frac{C}{A + B} \tag{55}$$

Notice that from the transition in equation 23:

$$\frac{A - B}{A + B} = tanh(\frac{\Delta|\delta r| - \ln(\frac{\sum_{a_- \in bad} P_\theta(a_-|q)}{\sum_{a_+ \in good} P_\theta(a_+|q)})}{2}) \tag{56}$$

Is the bound from theorem 1, and the second term:

$$\frac{C}{A + B} = \frac{\sum_{a \in misclassified} P_\theta(a|q)exp(M|\delta r|)}{\sum_{a_+ \in good} P_\theta(a_+|q) + \sum_{a_- \in bad} P_\theta(a_-|q)exp(-\Delta|\delta r|)} < \delta \cdot exp(M|\delta r|) \tag{57}$$

Lastly, notice that:

$$\frac{\sum_{a_- \in bad} P_\theta(a_-|q)}{\sum_{a_+ \in good} P_\theta(a_+|q)} = \frac{1 - B_0}{1 + B_0} \tag{58}$$

Where $B_0 = B[P_\theta(\cdot|q)]$, and that:

$$arctanh(B_0) = -\frac{1}{2} \ln \frac{1 - B_0}{1 + B_0} \tag{59}$$

Thus we obtain:

$$B[P_{\theta, r_e}(\cdot|q)] > tanh(\frac{\Delta|\delta r(q)|}{2} + arctanh(B_0)) - 2\delta \cdot exp(M|\delta r|) \tag{60}$$

Then, note that for coefficients that are not too large, $|\delta r(q)|$ is proportional to the injected vector coefficient $r_e$ (assumption 3), hence:

$$B[P_{\theta, r_e}(\cdot|q)] > tanh(\frac{\Delta\lambda}{2} \cdot r_e + arctanh(B_0)) - 2\delta \cdot exp(M\lambda \cdot r_e) \tag{61}$$

Where $\lambda$ is the coefficient relating $r_e$ to $|\delta r(q)|$. Thus for $r_e < \frac{\log \frac{\epsilon}{2\delta}}{M \cdot \lambda}$:

$$B[P_{\theta, r_e}(\cdot|q)] > tanh(\frac{\Delta\lambda}{2} \cdot r_e + arctanh(B_0)) - \epsilon \tag{62}$$

## G  RELATION OF REPRESENTATION ENGINEERING TO FINETUNING WITH PREFERENCE LEARNING

To a degree one can draw a relation between representation engineering and preference learning.

**Proposition 1** *For an LLM, one iteration of gradient descent on the preference learning loss with learning rate $\eta$ is equivalent to representation engineering with coefficient $r_e = \eta$.*

*Proof:*

The objective in preference learning is to minimize the loss:

$$L = -\mathbb{E}_{(x,y^+,y^-)\sim D}\big[\log \frac{P(y^+|x)}{P(y^-|x)}\big] = -\mathbb{E}_{(x,y^+,y^-)\sim D}[\langle r_x^{(L)}, U^T(e_{y_+} - e_{y_-})\rangle] \tag{63}$$

Which increases the likelihood of desired responses to prompts. By training with preference learning, in each iteration of gradient descent, each representation is changed by:

$$r^{(l)} \rightarrow r^{(l)} - \eta\frac{\partial L}{\partial r^{(l)}} \tag{64}$$

The gradient of the loss *w.r.t.* a hidden layer representation is:

$$\frac{\partial L}{\partial r^l} = \mathbb{E}_{(x,y^+,y^-)\sim D}\big[\frac{\partial r(x)}{\partial r^l(x)} \cdot U^T(e_{y_+} - e_{y_-})\big] \tag{65}$$

Thus at each layer, the representation is shifted in a direction that maximizes the difference between positive and negative responses' representations, $U^T(e_{y_+} - e_{y_-})$. Which is equivalent to representation engineering with coefficient $r_e = \eta$, and vectors $R_e = \{\mathbb{E}_{(x,y^+,y^-)\sim D}[\frac{\partial r(x)}{\partial r^l(x)} \cdot U^T(e_{y_+} - e_{y_-})]\}_{l=1}^L$

## H  EXTENSION OF RESULTS BEYOND BINARY BEHAVIOR SCORE

The idea behind theorem 1, is that the resulting change to the final hidden layer due to the representation injections linearly classifies aligned and misaligned answers, where the aligned/misaligned labels are given by the binary behavior scoring function. To extend beyond a binary behavior score, we need to assume that the model's latent space captures more finegrained differences between answers. Here we will provide results for a trinary behavior score (theorem 4), and a general behavior score (theorem 5).

A natural extension is for a trinary score function, where $\pm 1$ is aligned/misaligned, and $0$ is irrelevant/neutral. We can reformulate theorem 1 in the following way:

**Theorem 4** *Let $P_{\theta,r_e}(\cdot|q)$ be a model prompted with query $q$ and injected with representations of coefficient $r_e$. Let $B : \Sigma^* \rightarrow \{-1, 0, +1\}$ be a behavior scoring function. The injections to all layers amounts to a change in the final hidden layer representation that is $q$ dependent, denoted by the vector $\delta r_e^{(L)}(q)$. Assume that the representations of aligned and misaligned/irrelevant answers w.r.t. $B$ are linearly separable, and that $\delta r_e^{(L)}(q)$ linearly classifies them with margin $\Delta$. Then, the behavior expectation of the model conditioned on the query $q$ satisfies:*

$$B[P_{\theta,r_e}(\cdot|q)] \geq \frac{B_0 + P_+(e^{\Delta\lambda \cdot r_e} - 1)}{1 + P_+(e^{\Delta\lambda \cdot r_e} - 1)} \tag{66}$$

*Where $B_0 = B[P_\theta(\cdot|q)]$ and $P_+$ are the behavior expectation and probability of aligned answer without representation engineering, and $\lambda$ is a model dependent coefficient relating between $r_e$ and the corresponding final hidden state norm.*

The behavior bound has a different form, but it behaves the same – for $r_e = 0$, it coincides with $B_0$, around $r_e = 0$ it is linear, and for $r_e \rightarrow \infty$ it approaches $+1$. The proof, presented in H.1, essentially follows the proof of theorem 1, except besides the $P_\pm$ terms (probability mass of positive and negative responses without representation engineering) there is also a $P_0$ term.

For a general behavior scoring function, $B : \Sigma^* \to [-1, +1]$, we can similarly assume that the representations of answers with score $> b_+$ and answers with score $< b_+$, are linearly separable, and obtain the following result:

**Theorem 5** *Let $P_{\theta, r_e}(\cdot|q)$ be a model prompted with query $q$ and injected with representations of coefficient $r_e$. Let $B : \Sigma^* \to [-1, +1]$ be a behavior scoring function. The injections to all layers amounts to a change in the final hidden layer representation that is $q$ dependent, denoted by the vector $\delta r_e^{(L)}(q)$. Assume that the representations of answers with behavior score $> b+$ and those with score $< b_+$ w.r.t. $B$ are linearly separable, and that $\delta r_e^{(L)}(q)$ linearly classifies them with margin $\Delta$. Then, the behavior expectation of the model conditioned on the query $q$ satisfies:*

$$B[P_{\theta, r_e}(\cdot|q)] \geq \frac{b_+ P_+ e^{\Delta \lambda r_e} - P_-}{P_+ e^{\Delta \lambda r_e} + P_-} \tag{67}$$

*Where $P_{\pm}$ are the probabilities of aligned/misaligned answers without representation engineering, and $\lambda$ is a model dependent coefficient relating between $r_e$ and the corresponding final hidden state norm.*

Here we see that the behavior expectation converges to the maximal score $b_+$, for which $\delta r_e^{(L)}$ can classify answers below and above the score. The trend is similar to theorem 1, with a sigmoidal behavior, but without the tightness on behavior expectation at $r_e = 0$, due to the more complex behavior scoring function. The proof is presented in H.2.

## H.1 PROOF OF THEOREM 4

Following the same proof as in 1, up to equation 22, but replacing the sum over negative answers to sum over negative and neutral answers, we obtain by denoting $P_{\pm}$, the sum over positive/negative answers without representation engineering, and by $P_0$ sum over neutral answers:

$$B[P_{\theta, r_e}(\cdot|q)] \geq \frac{P_+ - P_- exp(-\Delta|\delta r|)}{P_+ + (P_- + P_0) exp(-\Delta|\delta r|)} \tag{68}$$

$$= \frac{P_+(e^{\Delta|\delta r|} - 1) + (P_+ - P_-)}{P_+(e^{\Delta|\delta r|} - 1) + (P_+ + P_- + P_0)} \tag{69}$$

We note that $P_+ + P_- + P_0 = 1$ and that $P_+ - P_- = B[P_{\theta, r_e=0}(\cdot|q)] = B_0$:

$$= \frac{P_+(e^{\Delta|\delta r|} - 1) + B_0}{P_+(e^{\Delta|\delta r|} - 1) + 1} \tag{70}$$

Lastly, applying assumption 3, replaces $|\delta r| = \lambda r_e$.

## H.2 PROOF OF THEOREM 5

Following the same proof idea as in theorem 1, starting with equation 12 but replacing the scores in the numerator for positive and negative answers with $b_+$ and $-1$ (for worst case), up to equation 22, denote by $P_+$ the probability without representation engineering for answers with score $> b_+$ and by $P_-$ the rest:

$$B[P_{\theta, r_e}(\cdot|q)] \geq \frac{b_+ P_+ e^{\Delta|\delta r|} - P_-}{P_+ e^{\Delta|\delta r|} + P_-} \tag{71}$$

Lastly, applying assumption 3, replaces $|\delta r| = \lambda r_e$.

# I EXTENSION OF RESULTS TO MULTI-TOKEN ANSWERS

Intuitively, both the alignment guarantee result of theorem 1 and helpfulness bound of theorem 2, which apply for a single token output, can be extended to multi-token answers by applying the results on multiple decoding steps.

## I.1 ALIGNMENT

Starting with alignment, we note that if the model is limited to producing $N$ tokens, then from corollary 1, we can ensure that with a large enough representation engineering coefficient, each token will correspond to an aligned response:

**Theorem 6** *Let $\epsilon > 0$, $P_\theta$ a language model, $B : \Sigma^* \to \{-1, +1\}$, behavior scoring function and $q$ a query, and suppose the model's reply contains at most $N$ tokens. Under the assumption of theorem 1 holding in every decoding step, for $r_e > \frac{1}{\Delta\lambda}(\log \frac{N}{\epsilon} + \log \frac{1-B_0}{1+B_0})$, then:*

$$B[P_\theta(\cdot|q)] > 1 - 2\epsilon \tag{72}$$

*Where $B_0$ is the behavior expectation without representation engineering.*

We see that larger coefficients of representation engineering improve the behavior expectation, similarly to corollary 1, but with multiple token answers. By inverting the relation between $r_e$ and $\epsilon$, and placing it in the behavior expectation bound, we obtain a sigmoid-like behavior, that is linear for $r_e \approx 0$.

*Proof:*

Following the notation of the proof of theorem 1, we note that at each decoding step, the probability of outputting a token $a_i$ that is aligned *w.r.t.* behavior scoring function $B$, conditioned on the previous context $qa_1...a_{i-1}$, is:

$$\frac{\sum_{a_+ \in good} P_{\theta,r_e}(a_+|qa_1...a_{i-1})}{\sum_{a_+ \in good} P_{\theta,r_e}(a_+|qa_1...a_{i-1}) + \sum_{a_- \in bad} P_{\theta,r_e}(a_-|qa_1...a_{i-1})} \tag{73}$$

Following the proof technique of theorem 1, we obtain that this probability is largest than:

$$\geq \frac{P_+ e^{\Delta\lambda r_e}}{P_+ e^{\Delta\lambda r_e} + P_-} \tag{74}$$

Where $P_\pm$ are the probabilities for an aligned/misaligned output at the given decoding step. To ensure this probability is larger than $1 - \epsilon'$, we demand:

$$r_e > \frac{\log \frac{P_-}{P_+} + \log \frac{1}{\epsilon'}}{\Delta\lambda} \tag{75}$$

Thus over $N$ decoding steps, we use a union bound, leading to a positive response with probability $(1-\epsilon')^N > (1 - \epsilon'N)$. Taking $\epsilon' = \epsilon/N$, we obtain:

$$r_e > \frac{\max_{i \in [N]}\{\log \frac{P_-^i}{P_+^i}\} + \log \frac{N}{\epsilon}}{\Delta\lambda} \tag{76}$$

Where $P_\pm^i$ is the probability for a positive/negative continuation in the $i$'th token of the response. We note that $\frac{P_-^i}{P_+^i} = \frac{1-B_0^i}{1+B_0^i}$, where $B_0^i$ is the behavior expectation at the $i$'th decoding step. For the response to be positive, it is required that every step is positive, due to the binary score, then the behavior expectation of the entire response is no larger than the behavior expectation of each decoding step, $B_0 \leq \min_{i \in [N]} B_0^i$, meaning it suffices to have:

$$r_e > \frac{\log \frac{1-B_0}{1+B_0} + \log \frac{N}{\epsilon}}{\Delta\lambda} \tag{77}$$

We obtain that under these conditions, an aligned response is generated with probability at least $1-\epsilon$. A negative response, is generated with probability no greater than $\epsilon$. Thus the behavior expectation is at least:

$$B[P_{\theta,r_e}(\cdot|q)] > 1 - 2\epsilon \tag{78}$$

## I.2 HELPFULNESS

For helpfulness, we will consider a query $q$ and a correct answer $a$ of $N$ tokens. We will show that the probability of the answer decreases quadratically. The intuition is that in each decoding step the probability decreases quadratically, and due to the probability chain rule, if at the $i$'th step of generation, the probability for the next token is $P_i$, then the full sequence probability is $\prod_{i=1}^N P_i$. Once we expand this term *w.r.t.* $r_e$, we get a leading quadratic dependence:

**Corollary 3** *Let $P_\theta$ be a language model and $q$ be a query with answer $a = a_1...a_N$ containing at most $N$ tokens. Denote by $\{P_0^i\}_{i=1}^N$ the probability assigned to each correct token $\{a_i\}_{i=1}^N$ in the sequence without representation engineering, such that the probability of the full sequence is $P_0 = \prod_{i=1}^N P_0^i$. Then under the conditions of theorem 2 holding at each decoding step, we have with probability of at least $1 - \frac{2N}{T}$:*

$$P_{\theta,r_e}(q) \leq \frac{P_0}{\prod_{i=1}^N (P_0^i + (1 - P_0^i)\alpha(1 - \epsilon)(1 + \frac{\lambda^2\sigma^2\beta^2}{2}r_e^2))} \tag{79}$$

This shows the original probability of the sequence $P_0$, is normalized by a term whose leading order is quadratic in $r_e$:

$$\prod_{i=1}^N (P_0^i + (1 - P_0^i)\alpha(1 - \epsilon)(1 + \frac{\lambda^2\sigma^2\beta^2}{2}r_e^2)) = \prod_{i=1}^N (P_0^i + (1 - P_0^i)\alpha(1 - \epsilon)) + c \cdot r_e^2 + o(r_e^2) \tag{80}$$

We once a gain note that if $P_0^i$ is close to 1, then $(P_0^i + (1 - P_0^i)\alpha(1 - \epsilon)) \approx 1$, making the bound tighter where the model is more helpful initially.

An alternative bound, is simply to consider that the probability for a sequence, $P_0$, is bounded by the probability of each element in the sequence, $P_0^i$, for which theorem 2 can be directly applied, and the quadratic decay is achieved, although this is a bound that is less tight.

