# OpenReview forum: "Tradeoffs Between Alignment and Helpfulness in Language Models with Representation Engineering"
_ICLR.cc/2025/Conference — ICLR 2025 Conference Withdrawn Submission_

### Official Review · Reviewer_sWNY · 2024-11-01

**Soundness:** 2
**Presentation:** 1
**Contribution:** 2
**Rating:** 3
**Confidence:** 3

**Summary:**

This paper presents a theoretical framework for analyzing the impact of representation engineering on Large Language Models (LLMs).  Representation engineering modifies the model's internal representations to achieve desired behaviors. This work examines how this technique affects both model helpfulness and safety alignment.

**Strengths:**

Representation editing is an emerging field of research, and developing theoretical foundations can accelerate progress in this area.
Establishing a theoretical framework to analyze the trade-off between evaluation metrics is highly valuable. In fact, a well-defined framework can extend beyond representation editing to inform a broader range of model optimization techniques.

**Weaknesses:**

The paper suffers from several shortcomings: The paper lacks clarity.  Crucial information such as a precise definition of the representation editing mechanism is omitted, while certain passages, like the assertion on Line 138 that "This is an accurate parameterization of state-of-the-art LLMs," offer no valuable insight and disrupt the flow of the paper.
Does the paper target static representation editing or dynamic editing? (please also refer to item 3)
The paper relies heavily on non-peer-reviewed research. It builds its theoretical framework upon these sources, raising concerns about the arguments' foundation. Examples include Zou et al. (2023a), Yan et al. (2024), and Jorgensen et al. (2023). This reliance makes it difficult to assess the validity of the assumptions, and the applicability of the theorems.
authors seem to focus on static editing (From lines 534-535 it seems to be static engineering), and their magnitude. Static editing: a fixed vector is injected throughout the auto-regressive text generation process. If my understanding is correct, in simple terms, the paper argues that injecting a vector in any direction will bias the softmax layer in LLM toward the tokens that are more closely aligned/associated with that vector, which is a known ML phenomenon. This significantly restricts the framework's contribution. (I’d like to see the author's counterarguments that bolster the contribution of the paper).
The paper's theoretical claims require stronger empirical support. For example, a figure illustrating the relationship between alignment manipulation and performance on TruthfulQA (or similar LLM benchmarks).

**Questions:**

Line 023: “First, we find that under the conditions of our framework, alignment can be guaranteed with representation engineering, and at the same time that helpfulness is harmed in the process.”
Alignment and helpfulness definition needs to be clearly defined in the paper to enable getting anything useful out of the proofs, i.e. does it mean that fairness score increase but output is a nonsense sequence of tokens? Is this called an aligned model? How is such a model useful?
Line 076: “with reduction from 50% success of adversarial attacks to less than 15%, and truthfulness enhancement, with a relative increase of over 50%, though at the cost of somewhat reducing the helpfulness of the model.”
Truthfulness and helpfulness distinction needs to be clarified in the paper. (1) there need to be definition for both metrics (with citations to peer review papers) to clarify the distinction in the context of LLM evaluation, (2) it needs to be discussed why the theoretical argument in this paper are applied to only helpfulness and not truthfulness (to be aligned with the aforementioned observation).
Line 215: In theorem 1, “Assume that the representations of aligned and misaligned answers w.r.t. B are linearly separable, and that δr(L) e (q) linearly classifies them with margin ∆.”
How do these assumptions affect the genericness and applicability of the theorem?
Line 80 “increasing body of works…” needs citations.

---

### Official Review · Reviewer_4qRJ · 2024-11-04

**Soundness:** 4
**Presentation:** 2
**Contribution:** 3
**Rating:** 6
**Confidence:** 3

**Summary:**

The paper proposes a theoretical framework to analyze how representation engineering on safety alignment would effect the ability of models to perform basic tasks. The paper provides bounds based on several assumptions in this senario. And adequate emperical results demonstrate that the framework works on downstream tasks using representation engineering.

**Strengths:**

+ This paper focus on a critical issue of tradeoffs between safety alignment and the general capabilities, when using representation engineering as inference time control.
+ This paper provides a theoretical framework based on several assumptions, and the assumptions are discussed in the appendix. Impressively, adequate emperical results also demonstrate the effectiveness of the framework on downstream tasks.
+ It's good to see a paper trying to give some solid theoratical analysis in a crowded research area. Recent trends also encourage some inference time work rather than different kinds of finetuning.

**Weaknesses:**

+ For the paper presentation, I'd like to see a seperate section for related works like model editing or jalibreak, instead of putting them in the introduction.
+ Although safety alignment is a major part of *Alignment* and it's good to study, I feel this paper doesn't cover the whole alignment senorias. Also, for the word *helpfulness*, it could be a bit different from general capabilities. Thus, the writing and the title of the paper may be polished to focus on what is actually studied.
+ As a theoretical framework, I expected to see a method or general idea on how to reach better **tradeoffs** as written in the title. However, most of the writing focuses on the analysis part. If seen as an analysis paper, I would expect more interesting results and claims to be here. For example, to find out why the models fail to answer correctly after representation engineering, e.g., comparing code solutions to Humaneval.

**Questions:**

+ Why on Humaneval has less experiments than on MMLU?
+ Why choose both `harmfulness` and `racism`? I think the later one is concluded.
+ misc: Line 469, 'as' should be capitalized

---

### Official Review · Reviewer_8zWf · 2024-11-04

**Soundness:** 2
**Presentation:** 2
**Contribution:** 2
**Rating:** 3
**Confidence:** 3

**Summary:**

The authors conduct a qualitative and quantitative analysis of how representation engineering impacts model alignment (harmlessness) and helpfulness (accuracy). Their findings suggest that alignment increases linearly with the norm of the representation engineering vector, while model helpfulness—defined as the likelihood of answering a query correctly—decreases quadratically with the vector norm.

The authors support their conclusions with substantial theoretical proofs, though their experimental validation could benefit from further refinement.

**Strengths:**

1. The authors thoroughly discuss the advantages and limitations of representation engineering, noting that it enhances robustness against adversarial attacks but at the cost of reduced helpfulness.
2. They provide a well-developed theoretical foundation to substantiate their conclusions.
3. Detailed experimental results are provided to illustrate their findings, with evaluations conducted on models from the Llama2 series.

**Weaknesses:**

1. The paper's focus seems to emphasize the trade-off between safety and capability. Helpfulness, however, is not solely about performance; it also involves instruction-following capabilities, among other aspects. Alignment, likewise, involves more than just safety; it also encompasses helpfulness and honesty [1].
2. The analysis centers on the Llama2 series, with two primary configurations: aligning an unaligned (pretrained) model and misaligning an aligned (RLHF) model. Expanding the scope to include other models could strengthen the generalizability of their claims.

[1] Bai, Y., Jones, A., Ndousse, K., Askell, A., Chen, A., DasSarma, N., & Kaplan, J. (2022). Training a helpful and harmless assistant with reinforcement learning from human feedback. arXiv preprint arXiv:2204.05862.

**Questions:**

1. Could the authors present accuracy variations on a wider range of datasets beyond MMLU and HumanEval, such as the MATH dataset? Expanding to other models could also enhance the robustness of the experiments.
2. Figures 4 and 5 would benefit from combined analysis, perhaps a quantitative extension of Figure 1. If the authors could visually represent the trend of decreasing helpfulness as safety increases, the trade-off would be more intuitive.

---

### Official Review · Reviewer_j4nM · 2024-11-11

**Soundness:** 2
**Presentation:** 3
**Contribution:** 2
**Rating:** 3
**Confidence:** 4

**Summary:**

The paper uses representation engineering as the alignment framework and explores tradeoffs between alignment and helpfulness on both theoretical and empirical aspects.  The following are the results.
1. The paper shows that alignment varies as an S-shaped curve and correctness varies as a bell-shaped curve as a function of engineered vectors.
2. The experiments validate S-curve and the bell curve. The model used is Llama2 13B model and the alignment performed is for racist-related harmfulness.

The findings provide quantitative basis on the tradeoffs.

**Strengths:**

1. Theoretical analysis of the tradeoffs between alignment and helpfulness is an important research topic.
2. The approach adopted in the paper is rigorous.

**Weaknesses:**

1. Representation engineering is not a mainstream approach for alignment. So the generalization of the approach to mainstream approaches like RLHF and RLAIF is not clear.

2. Helpfulness and alignment pertain to individual tasks. For example, the chat should be helpful and not harmful. Decoupling the evaluation reduces the realism of the experiments. Providing responses to HOWTO tasks, like creating a scientific experiment, but not to dangerous tasks like making explosives is an example of coupling the two types of tasks.

**Questions:**

1. How does the proposed approach apply to other alignment techniques like RLHF?
2. Is it possible to create synthetic datasets and show the effectiveness on a single task like HOWTO experiments?

---

### Note · Authors · 2024-11-30

I have read and agree with the venue's withdrawal policy on behalf of myself and my co-authors.